# On the role of entanglement and statistics in learning

**Srinivasan Arunachalam**
IBM Quantum
Almaden Research Center
Srinivasan.Arunachalam@ibm.com

**Vojtěch Havlíček**
IBM Quantum
T.J. Watson Research Center
Vojtech.Havlicek@ibm.com

**Louis Schatzki**
Electrical and Computer Engineering
University of Illinois, Urbana-Champaign
louisms2@illinois.edu

## Abstract

In this work we make progress in understanding the relationship between learning models when given access to entangled measurements, separable measurements and statistical measurements in the quantum statistical query (QSQ) model. To this end we prove the following results

1. For learning Boolean concept classes, we show that the entangled and separable sample complexity are *polynomially related*.

2. We give a concept class that shows an *exponential separation* between quantum PAC learning with classification noise and QSQ learning. This proves the "quantum analogue" of the seminal result of Blum et al. [1] that separates classical SQ learning from classical PAC learning with classification noise.

3. The main *technical contribution* is to introduce a quantum statistical query dimension (QSD), which we use to give lower bounds on the QSQ learning. Using this, we prove exponential QSQ lower bounds for testing purity of quantum states, shadow tomography, learning coset states for the Abelian hidden subgroup problem, degree-2 functions, planted bi-clique states and learning output states of Clifford circuits of depth $\mathrm{polylog}(n)$.

4. Using our QSQ lower bounds, we give an *unconditional* separation between weak and strong error mitigation and prove lower bounds for learning distributions in the QSQ model. Prior works by Quek et al. [2], Hinsche et al. [3] and Neitner et al. [4] proved the analogous results *assuming* diagonal measurements and our work removes this assumption.

## 1 Introduction

In the last few decades, machine learning (ML) has emerged as one of the most successful parts of artificial intelligence with wide-ranging applications in computer vision, image recognition, natural language processing. More recently, ML has been used in popular applications such as AlphaGo and Alpha zero (to play the games of Go and chess), chatGPT (to mimic a human conversation) and Alphafold (for solving some hard instances of protein folding). Simultaneously, understanding the power of quantum physics for ML has received much attention in the quantum computing community. There have already been many theoretical proposals for quantum algorithms providing speedups for practically relevant ML tasks such as clustering, recommendation systems, linear algebra, convex optimization, SVMs, kernel-based methods, topological data analysis [5–14]. There are several surveys dedicated to understanding the power of quantum methods for ML [15–18].

37th Conference on Neural Information Processing Systems (NeurIPS 2023).

Quantum learning theory provides a theoretical framework to understand quantum advantages in ML. Here, there is a concept class $\mathcal{C}$ which is a collection of $n$-qubit quantum states, a learner is provided with several copies of a state $\rho \in \mathcal{C}$, performs an arbitrary entangled operation on $\rho^{\otimes T}$ and the goal is to learn $\rho$ well-enough. This framework encompasses several results in quantum learning such as tomography, shadow tomography, learning interesting classes of states, learning an unknown distribution and functions encoded as a quantum state [19–26, 3, 2, 4, 17].

Given that machine learning is believed to be one of the first near-term applications of quantum computers, a natural question is how implementable are these algorithms in order to see quantum computational advantage in practice? The concern when considering near-term implementation of the above learning algorithms is twin-fold, $(i)$ the *infeasibilty* of preparing copies of $\rho$ and $(ii)$ performing arbitrary entangled measurements on many copies of $\rho$ at once, both of which seem out of reach for near-term devices. Motivated by near-term implementations, recently [27] introduced the model of *quantum statistical query* (QSQ) learning to understand the power of measurement statistics for learning, inspired by Kearns [28] classical SQ model. Since its introduction, variations of it have found applications in differential privacy, quantum supremacy, quantum neural networks, learning distributions and error mitigation [2–4, 29–31]. In the QSQ model, a learning algorithm can perform $\mathrm{poly}(n)$-many *efficiently*-implementable two-outcome measurements $\{M_i, \mathbb{I} - M_i\}$ and the goal is to learn the unknown $\rho$ well enough using the measurement statistics. Clearly this model is weaker than that given access to $\rho^{\otimes T}$, since the learner is only allowed access to *expectation values* over a single copy of $\rho$. Given that recent works on error mitigation have shown that good estimates of expectation values may be accessible well before fault-tolerant quantum computation [32, 33], understanding the power of QSQ algorithms is of fundamental importance.

In this work, we primarily consider concept classes constructed from Boolean functions. In Valiant's PAC learning framework, the goal is to learn a *concept class* $\mathcal{C} \subseteq \{c : \{0,1\}^n \to \{0,1\}\}$. In the PAC model,[1] a learning algorithm is given many uniformly random $(x^i, c^\star(x^i))$ where $c^\star \in \mathcal{C}$ is unknown and it uses these to learn $c^\star$ approximately well. Bshouty and Jackson introduced the *quantum* PAC (QPAC) model [34] wherein a quantum learner is given *quantum examples* $|\psi_{c^\star}\rangle^{\otimes T}$, i.e., coherent superpositions $|\psi_{c^\star}\rangle = \frac{1}{\sqrt{2^n}} \sum_x |x, c^\star(x)\rangle$, and it needs to learn the unknown $c^\star \in \mathcal{C}$ well enough. The complexity measure here is the *sample complexity*, i.e., copies of classical or quantum examples used by the algorithm. There have been works that have looked at this model and proven positive and negative results for learning function classes (see [17] for a survey).

**Relationships Between Models.** First, in the distribution-independent setting it is known that PAC and QPAC have the same sample complexity [35]. However, the same need not be true in the distribution-dependent setting, where, for example, quantum learners can efficient learn DNF formulas over the uniform distributions [34]. Second, it is a well-known fact that the classical *statistical query* (SQ) model is exponentially-separated from PAC learning as witnessed by the class of parity functions [28]. However, in [27] they show that parities can be learned in the QSQ model efficiently. In fact, in [27] they observed that, many positive results using quantum examples can be transformed into algorithms in the weaker QSQ framework. This motivates the following questions:

> *1. Do entangled measurements offer any advantages in learning function classes?*
>
> *2. Do measurement statistics suffice for learning function classes, i.e. is there a separation between* QSQ *and* QPAC*?*

Here, we resolve both these questions. We show that $(i)$ for learning *Boolean* function classes the sample complexity of learning with entangled measurements and separable measurements are polynomially related, thereby showing that separable measurements are sufficient to witness quantum speedups in practice and $(ii)$ there is an exponential separation between learning with entangled measurements (even in the presence of classification noise) and learning with just measurement statistics, thereby showing that just measurement statistics might be insufficient to witness quantum speedups in practice.

## 1.1 Main results

We now give a detailed description of our main results before we give an overview of their proofs in the next section.

---

[1]For simplicity, we discuss PAC learning under the *uniform*-distribution, i.e., $x$ is uniform in $\{0,1\}^n$.

**Entangled versus Separable measurements.** Since entangled measurements are vastly more difficult to realize experimentally, much recent work has gone into characterizing the limitiations of separable measurements. Bubeck et al. [36] gave a property testing task for which entangled measurements are *necessary* for obtaining the optimal bounds. More recently, for learning classes of arbitrary *quantum states* (i.e., not necessarily states constructed from function classes), two recent works by [26, 37] showed exponential separations for learning properties of quantum states with entangled vs separable measurements. Here, we study if similar separations exist when considering *function* classes, a small subset of all quantum states. Our first result shows that in order to exactly learn a function class, every learning algorithm using entangled measurements can be transformed into a learning algorithm using just separable measurements with a polynomial overhead in sample complexity.

**Result 1.** *For a concept class $\mathcal{C} \subseteq \{c : \{0,1\}^n \to \{0,1\}\}$, if $T$ copies of $|\psi_c\rangle$ suffice to learn an unknown $c \in \mathcal{C}$, then $O(nT^2)$ copies to learn $c$ using only separable measurements.*

**QSQ versus noisy-QPAC learning.** Classically, Kearns posed the question if SQ learning is equal to PAC learning with classification noise. The seminal result of Blum et al. [1] resolves this question by showing that the class of parity functions acting on the first $O((\log n) \cdot \log \log n)$ bits separates these two models of learning (under constant noise rate). In [27], the authors asked a question if there is a natural class of Boolean functions for which, QSQ learning can be separated from QPAC learning with noise (in fact, prior to our work, no separation was known even in the presence of *no noise*). Classically it is well-known that parities separates SQ learning from PAC learning. In [27], it was observed that the class of parities, juntas, and DNF formulas are learnable in the QSQ framework, leaning no clear candidate to separate QSQ from QPAC. This motivates the following questions:

($a$) In the noisy-quantum PAC model [34, 35], a learning algorithm is given copies of

$$|\psi_{c^\star}^n\rangle = \frac{1}{\sqrt{2^n}} \sum_{x \in \{0,1\}^n} |x\rangle\big(\sqrt{1-\eta}|c^\star(x)\rangle + \sqrt{\eta}|\overline{c^\star(x)}\rangle\big), \qquad (1)$$

and the goal is to learn $c^\star$. Is there a class that separates noisy-quantum PAC from QSQ learning?

($b$) Admittedly, the class constructed by Blum et al. [1] is "unnatural", can we obtain the separation in ($a$) for a *natural* concept class?

($c$) Does such a separation hold for non-constant error rate $\eta$?

Here, we describe a natural problem that witnesses this separation, resolving the three questions.

**Result 2.** *The concept class $\mathcal{C} = \{f_A : \{0,1\}^n \to \{0,1\} \mid f_A(x) = x^\top A x \ (mod\ 2), A \in \mathbb{F}_2^{n \times n}\}$ of all degree-$2$ Boolean functions can be (exact) learned using quantum examples and separable measurements even in the presence of $\eta$-classification noise in time $\mathrm{poly}(n, 1/(1-2\eta))$, whereas every QSQ algorithm requires $2^{\Omega(n)}$ queries to (approximately) learn $\mathcal{C}$.*

**Further applications.** While we first considered learning function classes, the QSQ model is meaningful for a much broader class of tasks in quantum information theory and to that end we prove the following: ($a$) we show hardness of shadow tomography (quadratically improving the prior bound [26] for separable measurements) and show hardness of even the simplest *Abelian* hidden subgroup problem in the QSQ model; ($b$) we give a *doubly*-exponential lower bound for testing purity of an unknown state; ($c$) we give an exponential separation between weak and strong error mitigation and ($d$) we give superpolynomial lower bounds for learning output distributions of quantum circuits (when measured in the computational basis). Prior works [2–4] considered tasks ($c$), ($d$) and showed these separations when the QSQ queries correspond to *diagonal* measurements, and we remove the diagonal assumption here. We discuss this further in Section 4.2.

## 2 Models of learning

We discuss the learning models of interest in our submission in this section. For a quick introduction to quantum information notation, we refer the reader to Section 1 in the Supplementary material. In all models, we are primarily interested in improper learning, i.e. the learner need not output a state or function from the concept class.

**Classical PAC learning.** Valiant [38] introduced the classical Probably Approximately Correct (PAC) learning model. In this model, a *concept class* $\mathcal{C} \subseteq \{c : \{0,1\}^n \to \{0,1\}\}$ is a collection of Boolean functions. The learning algorithm $\mathcal{A}$ obtains *labelled examples* $(x, c(x))$ where $x \in \{0,1\}^n$ is uniformly random and $c \in \mathcal{C}$ is the *unknown* target function.[2] The goal of an $(\varepsilon, \delta)$-learning algorithm $\mathcal{A}$ is the following: for every $c \in \mathcal{C}$, given labelled examples $\{(x^i, c(x^i))\}$, with probability $\geq 1 - \delta$ (over the randomness of the labelled examples and the internal randomness of the algorithm), outputs a succint circuit-representation for an hypothesis $h : \{0,1\}^n \to \{0,1\}$ such that $\Pr_x[c(x) = h(x)] \geq 1 - \varepsilon$. The sample complexity and time complexity of a learning algorithm is the maximal number of labelled examples and time used by the optimal learning algorithm respectively.

**Quantum PAC learning.** The quantum PAC model was introduced by Bshouty and Jackson [34] wherein, they allowed the learner access to quantum examples of the form $|\psi_c\rangle = \frac{1}{\sqrt{2^n}} \sum_{x \in \{0,1\}^n} |x, c(x)\rangle$. Like the classical complexities, one can similarly define the $(\varepsilon, \delta)$-sample and time complexity for learning $\mathcal{C}$ as the quantum sample complexity (i.e., number of quantum examples $|\psi_c\rangle$) used and quantum time complexity (i.e., number of quantum gates used in the algorithm) of an optimal $(\varepsilon, \delta)$-learner for $\mathcal{C}$. Similarly, Bshouty and Jackson [34] defined quantum learning with classification noise, wherein a learning algorithm is given access to $|\psi_c^n\rangle = \frac{1}{\sqrt{2^n}} \sum_x |x\rangle \otimes (\sqrt{1-\eta}|c(x)\rangle + \sqrt{\eta}|\overline{c(x)}\rangle)$. Such quantum examples have been investigated in prior works [34, 35, 39].

**Learning with entangled versus separable measurements.** Observe that in the usual definition of QPAC above, a learning algorithm is given access to $|\psi_c\rangle^{\otimes T}$ and needs to learn the unknown $c \in \mathcal{C}$. In this paper we make the distinction between the case where the learner uses entangled measurements, i.e., perform an arbitrary operation on copies of $|\psi_c\rangle$ versus the setting where the learner uses separable measurements, i.e., performs a single-copy measurement on every copy of $|\psi_c\rangle$ in the learning algorithm. When discussing learning with entangled and separable measurements, in this paper we will be concerned with *exact* learning, i.e., with probability $\geq 2/3$, the learner needs to *identify* $c$. We denote EntExact as the sample complexity of learning with entangled measurements and SepExact as the sample complexity of learning with separable measurements.

**Quantum statistical query learning.** We now discuss the QSQ model, following the definitions given in [27]. More generally, in order to learn an unknown quantum state $\rho$, in the QSQ model, the learner makes Qstat queries that take as input a bounded linear operator $M$ over the Hilbert space of $\rho$, satisfying $\|M\| \leq 1$, and tolerance $\tau$ and outputs a $\tau$-approximation of $\mathsf{Tr}(M\rho)$, i.e.,

$$\mathsf{Qstat} : (M, \tau) \mapsto \alpha \in [\mathsf{Tr}(M\rho) - \tau, \mathsf{Tr}(M\rho) + \tau].$$

The goal of the QSQ learner is: with probability $\geq 1 - \delta$, output a succinct description of a state $\sigma$ such that $\|\rho - \sigma\|_{\mathsf{Tr}} \leq \varepsilon$. In order to learn a function class using quantum examples, we have $\rho = |\psi_f\rangle\langle\psi_f|$ and on input $M$, the Qstat oracle responds with $\alpha \in [\langle\psi_f|M|\psi_f\rangle - \tau, \langle\psi_f|M|\psi_f\rangle + \tau]$.

In this case, the goal of a QSQ learner is to output a hypothesis $h$ that satisfies $d_{\mathsf{Tr}}(|\psi_f\rangle\langle\psi_f|, |\psi_h\rangle\langle\psi_h|) \leq \varepsilon$, which translates to $\Pr_x[h(x) = f(x)] \geq 1 - \sqrt{\varepsilon}$. The query complexity for learning $\mathcal{C}$, denoted $\mathsf{QSQ}(\mathcal{C})$, is the number of Qstat queries an optimal algorithm makes and the quantum time complexity is the total number of gates used by an optimal algorithm (which includes the gates to number of gates to implement $M$). We say a $n$-bit concept class $\mathcal{C}$ is QSQ learnable if $\mathcal{C}$ can be learned using $\mathrm{poly}(n)$ many Qstat queries, each with tolerance $\tau = 1/\mathrm{poly}(n)$ and observable $M$ which is implementable using $\mathrm{poly}(n)$ many gates. There are three ways to motivate the QSQ model

1. From a theoretical perspective, performing 2-outcome measurements is easier to implement than arbitrary separable measurements, which is in turn easier to implement than entangled measurements, so it is useful to understand the power of expectation values in quantum learning theory and the QSQ captures this question in a theoretical framework.[3]

---

[2]More generally in PAC learning, there is an unknown distribution $D : \{0,1\}^n \to [0,1]$ from which $x$ is drawn. Throughout this paper we will be concerned with uniform-distribution PAC learning, i.e., $D$ is the uniform distribution, so we describe the learning model for the uniform distribution for simplicity.

[3]Clearly any binary measurement $\{M, I - M\}$ can be simulated with a Qstat query to $M$ or $I - M$. In the opposite direction, any observable $M$ such that $\|M\| \leq 1$ can be converted into the POVM $\{\frac{I+M}{2}, \frac{I-M}{2}\}$. Thus, Qstat queries as defined above and binary POVMs are essentially the same model.

2. We emphasize that a QSQ learner is a *classical* algorithm since it receives statistical estimates of measurements on quantum states. One could envision a framework of learning where quantum states are prepared in the "cloud" and a *classical* learner needs to interact with the cloud only *classically*: QSQ models such a quantum framework.

3. Most quantum complexity classes are defined by making a binary measurement on a read-out qubit. This can be readily subsumed into the QSQ framework. The QSQ model also naturally extends recent works [2–4] wherein they consider the limitations of classical SQ in the setting where $M = \sum_x \phi(x)|x\rangle\langle x|$ is diagonal.

4. QSQ naturally generalizes SQ. That is, one can think of QSQ as being a form of statistical learning where the learner can change the basis of their statistics. Indeed, SQ corresponds to the case when all queried observables are diagonal.

**Some positive results in** QSQ    In quantum learning theory, there are a few well-known function classes that are learnable using quantum examples: parities, juntas, DNF formulas, the coupon collector problem, learning codeword states. It was observed in [35] that the first three classes are learnable in QSQ already, primarily because a version of Fourier sampling is implementable in QSQ. In this work we first observe that the coupon collector problem and learning codeword states are also learnable in the QSQ framework. We next observe that the class of Fourier-sparse functions are QSQ learnable (which subsumes all the positive results in [27]).

**Theorem 1.** *The class of $k$-Fourier sparse functions, the class of codeword states, coupon collector problem can be learned in the* QSQ *model.*

Beyond learning example states, we next observe that one can do tomography on the set of trivial states, i.e., states $|\psi\rangle = C|0^n\rangle$ where $C$ is a constant-depth $n$-qubit circuits, in polynomial time in the QSQ model. An open question of this work, and also the works of [3, 4], is if we can learn the distribution $P_C = \{\langle x|\psi\rangle^2\}_x$ using *classical* SQ queries. The theorem below shows that if we had direct access to $|\psi\rangle$, one can learn the state and the corresponding distribution $P_C$, using Qstat queries.

**Theorem 2.** *The class of $n$-qubit trivial states can be learned up to trace distance $\leq \varepsilon$ using* $\mathrm{poly}(n, 1/\varepsilon)$ Qstat *queries with tolerance* $\mathrm{poly}(\varepsilon/n)$.

For further details we defer the corresponding proofs to Section 5.1 in the Supplementary material.

## 3   Proof of results

In this section we outline the proof our two results.

### 3.1   Proof overview for Result 1

Our starting point towards proving this result is that one could use a result of Sen [40] that, given copies of $|\psi_{c^\star}\rangle$, one could apply *random* measurements on single copies of this state and produce an $h$ that is approximately close to $c^\star$ using at most $T = (\log|C|)/\varepsilon$ copies of $|\psi_{c^\star}\rangle$.[4] So, for separable learning, by picking $\varepsilon = \eta_{min}$ as the minimum distance between concepts in $\mathcal{C}$, one could exactly learn $\mathcal{C}$ using $T$ quantum examples. Proving a lower bound on entangled learning $\mathcal{C}$ is fairly straightforward as well: first observe that $(\log|\mathcal{C}|)/n$ is a lower bound on learning (since each quantum example gives $n$ bits of information and for exact learning one needs $\Omega(\log|\mathcal{C}|)$ bits of information) and also observe that $1/\eta_{min}$ is a lower bound, since to distinguish just between $c, c' \in \mathcal{C}$ that satisfy $\Pr_x[c(x) = c'(x)] = 1 - \eta_{min}$, so one needs $1/\eta_{min}$ copies of the unknown state. Putting this separable upper bound and entangled lower bound together gives us $\mathsf{SepExact}(\mathcal{C}) \leq n \cdot \mathsf{EntExact}(\mathcal{C})^2$ for all $\mathcal{C}$. This relation is however sub-optimal.

We further improve the entangled lower bound as follows. Let $\eta_a = \mathbb{E}_{c,c' \in \mathcal{C}} \Pr_x[c(x) \neq c'x)]$. We use a information-theoretic argument (inspired by a prior work [35]) as follows: define a random variable $\mathbf{A}, \mathbf{B}$ and the quantum state $\rho_{\mathbf{A},\mathbf{B}} = \sum_{c \in \mathcal{C}} |c\rangle\langle c| \otimes |\psi_c\rangle\langle\psi_c|^{\otimes k}$ (assuming that $k$ is the sample complexity of $\mathsf{EntExact}$). For exact learning we know that $I(\mathbf{A} : \mathbf{B}) = \Omega(\log|\mathcal{C}|)$ again because one needs to learn $\mathbf{A}$ exactly. Next we know that $I(\mathbf{A} : \mathbf{B}) \leq k \cdot I(\mathbf{A} : \mathbf{B}_1)$ (where $\mathbf{B}_1$ corresponds to the

---

[4]This idea was used in an earlier work of Chung and Lin [41] as well, but they weren't concerned with entangled and separable measurements.

first register in $\mathbf{B}_1$). Now using a non-trivial analysis, one can analyze the reduced density matrix on the subsystem $\mathbf{A}, \mathbf{B}_1$ and analyze its eigenvalues to show that $I(\mathbf{A} : \mathbf{B}_1) \leq n\eta_a$. Chaining these inequalities gives that $\mathsf{EntExact} \geq \max\{1/\eta_m, (\log |\mathcal{C}|)/(n\eta_a)\}$. Combining this entangled lower bound with the separable upper bound, we get that

$$\mathsf{SepExact} \leq O\Big(n \cdot \mathsf{EntExact} \cdot \min\big\{\eta_a/\eta_m \,,\, \mathsf{EntExact}\big\}\Big).$$

For further details we defer the proof to Section 2 in the Supplementary material. It is not hard to see that this relation is optimal as well for the class of degree-2 functions defined as

$$\mathcal{C} = \{f(x) = x^\top A x \pmod 2 : A \in \mathbb{F}_2^{n \times n}\}. \tag{2}$$

For this class $\eta_a = \eta_m = O(1)$ by the Schwartz-Zippel lemma. Recently it was shown [24] that $\mathsf{SepExact} = \Theta(n^2)$ and $\mathsf{EntExact} = \Theta(n)$ showing the optimality of our relation above.

### 3.2 Proof overview for Result 2

#### 3.2.1 Technical contribution.

A fundamental issue in proving our QSQ result is, what techniques could one use to prove these lower bounds? Prior to our work, in [27] they introduced two new techniques based on differential privacy and communication complexity that give lower bounds on QSQ complexity. However, both these lower bounds are exponentially weak! In particular, the lower bounds that they could prove were linear in $n$ for learning an $n$-bit concept class. Classically, there have been a sequence of works [42–44] with the goal of proving SQ lower bounds and finally the notion of *statistical dimension* was used to obtain close-to-optimal bounds for SQ learning certain concept classes and the breakthrough works of [44] used it to settle the complexity of learning the planted $k$-biclique distribution.

In this work, our technical contribution is a combinatorial parameter to lower bound QSQ complexity akin to the classical parameter. To this end, we follow a three-step approach.

1. **Reduction to Decision Problems.** We show that an algorithm $\mathcal{A}$ that learns a concept class below error $\varepsilon$ in trace distance using Qstat queries of tolerance $\tau$ can also be used to solve the following decision problem: for a fixed $\sigma$ such that $\min_{\rho \in \mathcal{C}} d_{\mathsf{Tr}}(\rho, \sigma) > 2(\tau + \varepsilon)$, decide if an unknown state is either some $\rho \in \mathcal{C}$ or equals $\sigma$. Calling QQC the complexity of such decision problem, we show that

$$\mathsf{QSQ}(\mathcal{C}) \geq \max_\sigma \{\mathsf{QQC}(\mathcal{C}, \sigma) - 1 : \min_{\rho \in \mathcal{C}} d_{\mathsf{Tr}}(\rho, \sigma) > 2(\tau + \varepsilon)\}.$$

2. **Quantum Statistical Dimension.** Next, we define the notion of *quantum statistical dimension* QSD: for $\tau > 0$, a class of states $\mathcal{C}$ and a $\sigma \notin \mathcal{C}$, the $\mathsf{QSD}_\tau(\mathcal{C}, \sigma)$ is the smallest integer such that there exists a distribution $\nu$ over Qstat queries $M$ satisfying $\Pr_{M \sim \nu}[|\mathsf{Tr}(M(\rho - \sigma))| > \tau] \geq 1/d$ for all $\rho \in \mathcal{C}$. From an operational perspective QSD is natural, as it can be viewed as the smallest expected number of observables that can distinguish all states in $\mathcal{C}$ from $\sigma$. We then show that if the decision algorithm succeeds with probability at least $1 - \delta$, we have that:

$$\mathsf{QQC}(\mathcal{C}, \sigma) \geq (1 - 2\delta)\mathsf{QSD}_\tau(\mathcal{C}, \sigma).$$

3. **Lower Bounds on QSD.** Even with this lower bound, proving bounds on $\mathsf{QSD}(\mathcal{C}, \sigma)$ is non-trivial. To this end, we further give two lower bounding techniques for $\mathsf{QSD}(\mathcal{C}, \sigma)$, one based on the variance of Qstat queries across $\mathcal{C}$ (inspired by the work of Kearns [28]) and one based on average correlation (inspired by the work of Feldman [43]). In particular, we define two combinatorial quantities $\mathsf{Var}(\mathcal{C})$ and $\mathsf{QAC}(\mathcal{C}, \sigma)$ which can be associated with every class and use it to lower bound QSD.

   Let $\mu$ be a distribution over $\mathcal{C}$, such that $\sigma_1 := \mathbb{E}_{\rho \sim \mu}[\rho] \notin \mathcal{C}$. We define $\mathsf{Var}(\mathcal{C})$ as follows:

$$\mathsf{Var}(\mathcal{C}) = \tau^2 \cdot \min_{M, \|M\| \leq 1} \Big(\mathsf{Var}_{\rho \sim \mu}[\mathsf{Tr}[\rho M]]\Big)^{-1}, \tag{3}$$

   where

$$\mathsf{Var}_{\rho \sim \mu}[\mathsf{Tr}(\rho M)] = \mathbb{E}_{\rho \sim \mu}[\mathsf{Tr}(\rho M)^2] - \Big(\mathbb{E}_{\rho \sim \mu}[\mathsf{Tr}(\rho M)]\Big)^2.$$

We show that $\mathsf{QSD}_\tau(\mathcal{C}, \sigma_1) \geq \mathsf{Var}(\mathcal{C})$, which we eventually use to lower bound $\mathsf{QSQ}(\mathcal{C})$. Next, for $\sigma_2 \notin \mathcal{C}$, we define the average correlation $\mathsf{QAC}(\mathcal{C}, \sigma_2)$ as

$$\mathsf{QAC}(\mathcal{C}, \sigma_2) = \sup_{\mathcal{C}_0 \subset \mathcal{C}} \left( \kappa_\tau^\gamma\text{-}\mathsf{frac}(\mathcal{C}_0, \sigma_2) \right)^{-1}, \tag{4}$$

where $\kappa_\tau^\gamma\text{-}\mathsf{frac}(\mathcal{C}_0, \sigma_2)$ is a combinatorial parameter capturing correlations between states in $\mathcal{C}_0$ and $\sigma_2$. We then show that $\mathsf{QSD}_\tau(\mathcal{C}, \sigma_2) \geq \mathsf{QAC}(\mathcal{C}, \sigma_2)$ and use this in turn to lower bound $\mathsf{QSQ}(\mathcal{C})$.[5]

Putting together the three bullets above, the QSQ complexity of learning can be lower bounded by the variance bound and the average correlation bound that we define in this work. We remark that although, our quantum combinatorial parameters are inspired by the classical works of Feldman et al. [42–44], proving that they lower bound QSQ complexity and also giving lower bounds for the corresponding concept class using these parameters is non-trivial and is a key technical contribution of our work. Below, we apply these lower bounds to obtain our learning results. For further details we defer the formal definitions and proofs to Section 3 in the Supplementary material.

### 3.2.2 QSQ **versus noisy** QPAC

We now sketch the proof of Result 2. As mentioned earlier, previous to our work we did not have a candidate class to separate QPAC from QSQ (let alone with noise). There have been a few works that have shown exponential lower bounds for learning using separable measurements [26, 37, 45], but all these lower bounds correspond to learning classes of mixed quantum states. Hence it was open if there is very simple structured *function class* such that quantum examples corresponding to this function class is hard for QSQ (in fact given our polynomial relation between entangled and separable learning, it is conceivable that for the small class of function states, QSQ are QPAC are polynomially related as well). In this work, we look at the degree-2 concept class $\mathcal{C}$ defined in Eq. (2).

Recently, it was observed that [24] this class is learnable using $O(n)$ quantum examples with entangled measurements and $O(n^2)$ quantum examples with separable measurements. Our main contribution is in showing that the QSQ complexity of learning $\mathcal{C}$ with tolerance $\tau$ is $\Omega(2^{n/2} \cdot \tau^2)$. When $\tau = 1/\operatorname{poly}(n)$ this is an exponential, $2^{\Omega(n)}$, lower bound. We prove the hardness for algorithms using Qstat queries using the variance lower bounding technique. In particular, we show that for every $n+1$ qubit operator $M$ such that $\|M\| \leq 1$, we have that

$$\mathsf{Var}_f(\mathsf{Tr}[M\psi_f]) = O(2^{-n/2}), \tag{5}$$

where we let $\psi_f = |\psi_f\rangle\langle\psi_f|$ for notational simplicity. Combined with our variance lower bound introduced in Section 3.2.1, we obtain our lower bound on the QSQ complexity of learning $\mathcal{C}$. It remains to establish Eq. (5). To this end, we need to understand

$$\mathsf{Var}_f(\mathsf{Tr}[M\psi_f]) = \mathbb{E}_f[\mathsf{Tr}[M\psi_f]^2] - (\mathbb{E}_f[\mathsf{Tr}[M\psi_f]])^2 \tag{6}$$

To do so, we decompose $\psi_f$ as follows. For every $f : \{0,1\}^n \to \{0,1\}$ let $|\psi_f\rangle = \frac{1}{\sqrt{2^n}} \sum_x |x, f(x)\rangle$ and $|\phi_f\rangle = \sum_x (-1)^{f(x)} |x\rangle$. For convenience we let $|u\rangle = \frac{1}{\sqrt{2^n}} \sum_x |x\rangle$. Then we see that

$$|\psi_f\rangle\langle\psi_f| = \frac{1}{2} \Big( \underbrace{|\phi_f\rangle\langle\phi_f| \otimes |-\rangle\langle-|}_{\rho_1^f} - \underbrace{|\phi_f\rangle\langle u| \otimes |-\rangle\langle+|}_{\rho_2^f} - \underbrace{|u\rangle\langle\phi_f| \otimes |+\rangle\langle-|}_{\rho_3^f} + \underbrace{|u\rangle\langle u| \otimes |+\rangle\langle+|}_{\rho_4^f} \Big).$$

We now note that any $n+1$ qubit observable $M$ can be decomposed as $M = \sum_{a,b} M_{a,b} \otimes |a\rangle\langle b|$ where now $a, b \in \{+, -\}$. Since $\|M\| \leq 1$ we also have that $\|M_{a,b}\| \leq 1$, however the off-diagonal blocks now no longer need be Hermitian. In an abuse of notation we now discard the last qubit of $\rho_i^f$ and denote the resulting state also as $\rho_i^f$. For ease of notation we further introduce the notation $M_1 = M_{-,-}$, $M_2 = M_{-,+}$, $M_3 = M_{+,-}$, and $M_4 = M_{+,+}$ Thus, we see that $\mathsf{Tr}[M\psi_f] = \frac{1}{2} \sum_i \mathsf{Tr}[M_i \rho_i^f]$ and further the variance can be written as

---

[5]We point out two subtleties, discussed in detail in Section 3 of the Supplementary Material. Firstly, our definition of QAC is defined only for $\sigma$s that are full rank. Secondly, as stated, the $\mathsf{Var}(\mathcal{C})$ lower bound only yields a QSQ lower bound if $\min_{\rho \in \mathcal{C}} d_{\mathsf{Tr}}(\rho, \sigma_1) > 2(\tau + \varepsilon)$. This does not impact the lower bound proofs by this technique that we present here. In Section 3 of the supplementary material, we also briefly discuss an alternative approach to hiding decision problems in $\mathcal{C}$ that relaxes this condition.

$$\text{Var}_f(\text{Tr}[M\psi_f]) = \frac{1}{4} \sum_{i,j} \left[ \mathbb{E}_f \text{Tr}(M_i \rho_i^f) \cdot \text{Tr}(M_j \rho_j^f) - \text{Tr}(M_i \mathbb{E}_f[\rho_i^f]) \cdot \text{Tr}(M_j \mathbb{E}_f[\rho_j^f]) \right]. \quad (7)$$

At this point, we upper bound all these terms by $\exp(-n/2)$. Proving this upper bound is fairly combinatorial but crucially it involves understanding the properties of the ensemble $\{|\psi_f\rangle\}_f$ and its moments for a uniformly random degree-2 functions $f$. Finally, we observe that the concept class can be learned given noisy quantum examples like in Eq. (1) using $\text{poly}(n, 1/(1-2\eta))$ examples (proving this uses the standard procedure to take derivatives of quantum states and the observation that this procedure is noise-resilient). This gives us the claimed separation between QSQ and noisy-QPAC, the "quantum analogue" of the seminal result of Blum et al. [1] for a *natural* class and with *non-constant* error rate close to $1/2$. We refer the interested reader to Section 4.1 in the Supplementary material.

### 3.3 Smallest class separation

Above we saw that the concept class of quadratic functions separated QPAC from QSQ. Observe that states in this concept class can be prepared by circuits of size $O(n^2)$ and depth $O(n)$ consisting of $\{\text{Had}, \text{X}, \text{CX}\}$ gates. A natural question is, can states prepared by *smaller* circuits also witness such a separation between QPAC and QSQ? In Section 2 we saw that trivial states are learnable in QSQ, so is it necessary to have super-constant depth in order to show super-polynomial lower bounds? In the theorem below we answer this in the positive, by using a simple padding argument inspired by a prior work of Hinshe et al. [3]. In particular, we show that the class of states produced by $\omega(\log n)$-depth is already hard to learn for QSQ algorithms using a polynomial number of queries. For further details we defer the proof to Section 4.2 in the Supplementary material.

**Theorem 3.** *For any $\alpha \in (0,1)$ there exists a family of $n$ qubit Clifford circuits of depth $d = (\log n)^{1/\alpha}$ and size $d^2$ that requires $2^{\Omega(d)}$ Qstat queries to learn the state to error $\leq 1/2$ in trace distance.*

## 4 Further applications

In this section we first use our lower bound techniques to give QSQ lower bounds for interesting classes of states and then use our QSQ lower bounds for further applications outside learning. For further details we defer the proof to Sections 4 and 5 in the Supplementary material.

### 4.1 QSQ lower bounds for learning states

Extending our lower bounds from above, we consider fundamental problems in quantum computing and prove QSQ lower bounds for these tasks. We summarize our results in the table below, for $\tau = O(\frac{1}{\text{poly}(n)})$, before expanding upon these in the following subsections.

| Problem | QSQ Bound | General Complexity |
|---------|-----------|--------------------|
| Shadow tomography with Pauli observables | $\Omega(4^n)$ | $O(2^n)$ with *separable* measurements [26] |
| Learning coset states from Abelian hidden subgroup problem | $2^{\Omega(n)}$ | $O(n)$ with *separable* measurements |
| Purity testing | $2^{2^{\Omega(n)}}$ | $O(1)$ with *entangled* measurements $\Theta(2^n)$ with *separable* measurements [26] |
| Learning states from $O(2^n)$ approximate 2-designs | $2^{\Omega(n)}$ | $O(n)$ for stabilizer states with *entangled* measurements |

*Hidden subgroup problem.* Coset states appear often in the hidden subgroup problem (HSP) [46, 47], a fundamental problem in quantum computing. It is well-known that coset states of the Abelian HSP can be learned exactly from *separable* measurements in polynomial sample complexity and for non-Abelian groups, it was well-known that separable measurements [48, 49] require *exponential* many copies to learn a coset state. A natural question is, what is the QSQ complexity of learning coset states? Given that the *standard approach* for HSP is based of Fourier sampling and [27] showed that a version of Fourier sampling is easy in QSQ, it is natural to expect that HSP is implementable in QSQ. Surprisingly, in this work, we show that, even for *Abelian* groups, the QSQ sample complexity of learning the unknown coset state is exponentially large. In particular, we show a lower bound of

$\Omega(\tau^4 \cdot 2^n)$ on the QSQ complexity of learning using $\mathsf{Qstat}(\tau)$ queries and the proof of this is done via the average correlation method we introduced in Section 3.2.1.

***Shadow tomography.*** In recent times, there have been a lot of works surrounding the framework of shadow tomography [21, 37]. The goal here is, given copies of an unknown quantum state $\rho$, the learner has to predict the expectation value $\mathsf{Tr}[O_i\rho]$ of a collection of known observables $\{O_i\}_{i\in[k]}$ up to error $\varepsilon$. It is well-known to be solvable using $\mathrm{poly}(n, \log k)$ copies of $\rho$. In [26] the authors show $\Theta(2^n)$ copies of $\rho$ are necessary and sufficient for shadow tomography using *separable measurements*. To prove the lower bounds the authors construct a many-vs-one decision task where $\sigma = \mathbb{I}/2^n$ and $\mathcal{C} = \{\rho_i = \frac{\mathbb{I}+3\varepsilon O_i}{2^n}\}$. Assuming that $\mathsf{Tr}[O_i] = 0$ and $\mathsf{Tr}[O_i^2] = 2^n$ for all $O_i$, then an algorithm which solves the shadow tomography problem with high probability also solves the decision problem. Thus, a lower bound on the latter is also a lower bound on the sample complexity of shadow tomography. Here we give a quadratically *stronger* lower bound of $\Omega(4^n)$ when given access to only $\mathsf{Qstat}$ measurements, which we prove using the average correlation method. Our result shows that even separable measurements and not just statistics play a non-trivial role in shadow tomography.

***Does tolerance matter?*** A natural question when discussing QSQ learning is, is there a natural *distribution learning* task that can be solved with tolerance $\tau_Q \geq \tau_C$ such that classical $\mathsf{Stat}(\tau_C)$ queries cannot solve the task but $\mathsf{Qstat}(\tau_Q)$ can solve the task? Here we consider the class of *bi-clique states* introduced in the seminal work of Feldman et al. [44]. In their work they showed that for detecting a planted bipartite $k$-clique distributions when the planted $k$-clique has size $n^{1/2-\varepsilon}$ (for constant $\varepsilon > 0$), it is necessary and sufficient to make superpolynomial in $n$ many $\mathsf{Stat}(k/n)$ queries. Here we show that one can achieve the same query complexity quantumly but with $\mathsf{Qstat}(\sqrt{k/n})$, i.e., with quadratically larger tolerance we can detect a $k$-biclique. A classical $\mathsf{SQ}$ algorithm cannot solve this task with $\tau_C = \sqrt{k/n}$ queries.

***A doubly exponential lower bound?*** So far all our lower bounds for learning $n$-qubit quantum states are exponential in $n$. A natural question is, can one prove a doubly exponential lower bound for some task? In this work, we show that the natural problem of *testing purity*, i.e., given a quantum state $\rho$ return an estimate of $\mathsf{Tr}[\rho^2]$, requires $\exp(2^n\tau^2)$ many QSQ queries to solve. Previous work of [26] showed that it is necessary and sufficient to use $\Theta(2^n)$ many copies of $\rho$ to test purity if we were allowed *separable* measurements, but our work considers the weaker QSQ model and proves a doubly-exponential lower bound. The proof of this uses Levy's lemma and the ensemble of Haar random states to lower bound the quantum statistical dimension in a manner similar to that of the variance based technique.

## 4.2 Applications outside learning

Using our QSQ results we present two applications. First we give an exponential separation between weak and strong error mitigation, resolving an open question of Quek et al. [2]. Second, we show super-polynomial lower bounds for learning output distributions (in the computational basis) of $n$-qubit Clifford circuits of depth $\omega(\log n)$ and Haar random circuit of depth-$O(n)$, extending the works of [3, 4]. All these results [2–4] proved these lower bounds for QSQ algorithms assuming *diagonal* observables. For further details we defer the proof to Section 6 in the Supplementary material.

***Error mitigation.*** Error mitigation (EM) was introduced as an algorithmic technique to reduce the noise-induced in near-term quantum devices, hopefully with a small overhead, in comparison to building a full-scale fault-tolerant quantum computer to harness general quantum advantages [50]. In recent times, EM has obtained a lot of attention with several works understanding how to obtain *near-term* quantum speedups as a surrogate to performing error correction. EM has been an important component in recent QML demonstrations [37, 51, 10].

More formally, an EM algorithm $\mathcal{A}$ takes as input a quantum circuit $C$, noise channel $\mathcal{N}$ and copies of $|\psi'\rangle = \mathcal{N}(C)|0^n\rangle$. In a strong EM protocol, $\mathcal{A}$ needs to produce samples from a distribution $D$ that satisfies $d_{\mathsf{TV}}(D, \{\langle x|C|0^n\rangle^2\}_x) \leq \varepsilon$ and in the weak EM setting, given observables $M_1, \ldots, M_k$ the goal is to approximate $\langle\psi|M_i|\psi\rangle$ upto $\varepsilon$-error. In [2], they asked the question: how large should $k$ be in order to simulate weak EM by strong EM? They show that *when $M_i$s are diagonal*, then $k = \Omega(2^n)$, i.e., they gave an exponential separation between weak and strong EM. In this work, our main contribution is to use Result 2 to remove the assumption that $M_i$s are diagonal and show an exponential separation unconditionally between weak and strong EM.

***Learning distributions.*** Recently, the works of Hinsche et al. [3] and Nietner et al. [4] initiated the study of learning output distributions of quantum circuits. In particular, they considered the following *general* question: Let $|\psi_U\rangle = U|0^n\rangle$ where $U \in \mathcal{U}$ is a family of interesting unitaries and let $P_U(x) = \langle x|U|0^n\rangle^2$ be a distribution. How many QSQ queries does one need to learn the $P_U$ to $d_{\mathsf{TV}}$ at most $\varepsilon$? To this end, the works of [3, 4] looked at *diagonal $M$s*, i.e., $M = \sum_X \phi(x)|x\rangle\langle x|$ for $\phi : \{0,1\}^n \to [-1,1]$ and showed the hardness of approximately learning $P_U$ for $\mathcal{U}$ being $\omega(\log n)$-depth Clifford circuits and depth-$d \in \{\omega(\log n), O(n)\}$ and $d \to \infty$-depth Haar random circuits.[6]

In this work, we improve upon their lower bounds by removing the assumption that $M$ is diagonal and prove a *general* QSQ lower bounds for these circuit families that is considered in their work. In order to prove this bound, we follow the following three step approach ($i$) We first observe the following simple fact: for distributions $p, q : \mathcal{Z} \to [0, 1]$, define $|\psi_p\rangle = \sum_{z \in \mathcal{Z}} \sqrt{p(z)}|z\rangle$ and $|\psi_q\rangle$ similarly. Then $\||\psi_p\rangle - |\psi_q\rangle\|_{\mathsf{Tr}}^2 \leq 2\|p - q\|_{tvd}$. So it suffices to prove the hardness of learning the output state, in order to prove the hardness of learning the distribution. ($ii$) Next we consider the class $\mathcal{C}$ of states forming a $\gamma$-approximate $2$−design where $\gamma = O(2^{-n})$ and show that learning states from $\mathcal{C}$ with error $\leq 2/3$ in trace distance requires $\Omega(\tau^2 \cdot 2^n)$ $\mathsf{Qstat}(\tau)$ queries. ($iii$) Finally, using that depth-$O(n)$ circuits form a design, we invoke our QSQ lower bound in order to prove the hardness of learning the output states of these circuits. These three steps proves the hardness of learning output distributions (in the computational basis) of quantum circuits.

**Open questions.** There are a few natural questions that our work opens up: ($i$) Can we show that for every concept class $\mathcal{C}$, we have that $\mathsf{SepExact} \leq O(n \cdot \mathsf{EntExact})$?, ($ii$) Following [3, 4] what is QSQ complexity of learning the output distribution of constant-depth circuits assuming we only use *diagonal* operators? ($iii$) Theoretically our work separates weak and strong error mitigation, but in practice there are often assumptions in the mitigation protocols, can we show theoretical separations even after making these assumptions? ($iv$) Classically it is well-known that several algorithms can be cast into the QSQ framework, is the same true quantumly? If so, that would suggest that QSQ as a unifying framework for designing new learning algorithms. ($v$) What is the QSQ complexity of the Hidden subgroup problem when given access to *function* states, instead of coset states (which is the case only in the *standard approach*).

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
