# On the role of entanglement and statistics in learning (Supplementary material)

**Srinivasan Arunachalam**
IBM Quantum
Almaden Research Center
Srinivasan.Arunachalam@ibm.com

**Vojtěch Havlíček**
IBM Quantum
T.J. Watson Research Center
Vojtech.Havlicek@ibm.com

**Louis Schatzki**
Electrical and Computer Engineering
University of Illinois, Urbana-Champaign
louisms2@illinois.edu

## 1 Preliminaries

### 1.1 Quantum Information Theory

Qubits are unit vectors in $\mathbb{C}^2$ with a canonical basis given as $|0\rangle = \begin{pmatrix} 1 \\ 0 \end{pmatrix}$ and $|1\rangle = \begin{pmatrix} 0 \\ 1 \end{pmatrix}$. Pure quantum states composed of $n$ qubits are unit vectors in $\mathbb{C}^{2^n}$. Following Dirac notation we indicate a state via $|\psi\rangle$ and its conjugate transpose, an element of $(\mathbb{C}^{2^n})^*$, by $\langle\psi|$. Outer products are indicated by the notation $|\psi\rangle\langle\phi|$. Mixed states/density matrices $\rho$ are positive semi-definite operators on $\mathbb{C}^{2^n}$ such that $\text{Tr}[\rho] = 1$. Any mixed state can be decomposed into a probability distribution over projectors onto pure states $\rho = \sum_i \lambda_i |u_i\rangle\langle u_i|$ where $\sum_i \lambda_i = 1$. Pure states correspond to rank 1 mixed states. We will often label the mixed state corresponding to a pure state $|\psi\rangle$ by $\psi$ (instead of $|\psi\rangle\langle\psi|$). Note that this is defined *up to* an absolute phase, i.e. $|\psi\rangle$ and $e^{i\theta}|\psi\rangle$ correspond to the same density matrix for all $\theta \in \mathbb{R}$. Positive operator valued measures (POVMs) capture the most general notion of quantum measurements. These are given by ensembles of positive semi-definite operators $\{E_i\}_i$ such that $\sum_i E_i = \mathbb{I}$. The probability of measurement outcome $i$ is given by $\text{Tr}[E_i\rho]$. Observables $M$ are bounded Hermitian operators on $\mathbb{C}^{2^n}$, representing a measurement with values assigned to the outcomes. The expectation value of an observable is given by $\text{Tr}[M\rho]$. Quantum computers operate by applying gates (unitary matrices) to (ideally) pure states which evolve like $|\psi'\rangle = U|\psi\rangle$ where $U$ is some unitary. One important gate we will see is the Hadamard gate given by $H = \frac{1}{\sqrt{2}} \begin{pmatrix} 1 & 1 \\ 1 & -1 \end{pmatrix}$.

### 1.2 Notation

Throughout for $n \geq 1$, we let $[n] = \{1, \dots, n\}$. For quantum states $\rho, \sigma$, we denote $d_{\text{Tr}}(\rho, \sigma)$ as the trace distance between the states $|\phi\rangle$ and $|\psi\rangle$, defined as

$$d_{\text{Tr}}(\rho, \sigma) = \frac{1}{2}\|\rho - \sigma\|_1 \,,$$

where $\|\cdot\|_1$ is the Schatten-1 norm of a matrix. For pure states $|\phi\rangle$ and $|\psi\rangle$ this simplies to

$$d_{\text{Tr}}(|\phi\rangle, |\psi\rangle) = \sqrt{1 - |\langle\phi|\psi\rangle|^2} \,.$$

For distributions, $P, Q : \{0, 1\}^n \to [0, 1]$, we say $\text{Pr}_{x \sim P}$ to mean $x$ is sampled from $P$. We indicate sampling from the uniform distribution on a set $\mathcal{C}$ by $x \sim \mathcal{C}$. Similarly, by $d_{\text{TV}}(P, Q)$ we mean the

37th Conference on Neural Information Processing Systems (NeurIPS 2023).

total-variational distance between $P, Q$. Similarly, define the Hellinger distance between $P, Q$ as

$$d_{\mathsf{H}}(P,Q)^2 = 1 - \left( \sum_x \sqrt{P(x)Q(x)} \right)^2.$$

## 1.3 Useful theorems

**Theorem 1** ([1] (paraphrased)). *Given two distinct pure states $|\psi\rangle$ and $|\phi\rangle$ in $\mathbb{C}^{2^n}$ there exists a constant $k$ such that if $2 < k2^{n-1}$ then with probability at least $1 - \exp(-k2^{n-1})$ over random bases $M$ we have that $\|M(|\psi\rangle) - M(|\phi\rangle)\|_1 > \sqrt{2}k d_{\mathsf{Tr}}(|\phi\rangle, |\psi\rangle)$, where $M(\sigma)$ denotes the probability distribution induced by measuring the basis $M$ on state $\sigma$.*

**Theorem 2** (Levy's lemma). *Let $f : S^{d-1} \to \mathbb{C}$ be a function on the $d$-dimensional unit sphere $S^{d-1}$. Let $k$ be such that for every $|\phi\rangle, |\psi\rangle \in S^{d-1}$, we have that*

$$|f(|\psi\rangle) - f(|\phi\rangle)| \le k \cdot \|\phi - \psi\|_2,$$

*then there exists a constant $C > 1$ such that*

$$\Pr\left[ |f(\psi) - \mathbb{E}[f(\psi)]| \ge \varepsilon \right] \le 2\exp(-Cd\varepsilon^2/k^2),$$

*where the probability and expectations are over the Haar measure on the sphere.*

**Fact 1.** *Let binary random variable $\mathbf{b} \in \{0,1\}$ be uniformly distributed. Suppose an algorithm is given $|\psi_{\mathbf{b}}\rangle$ (for unknown b) and is required to guess whether $\mathbf{b} = 0$ or $\mathbf{b} = 1$. It will guess correctly with probability at most $\frac{1}{2} + \frac{1}{2}\sqrt{1 - |\langle\psi_0|\psi_1\rangle|^2}$.*

*Note that if we can distinguish $|\psi_0\rangle, |\psi_1\rangle$ with probability $\ge 1 - \delta$, then $|\langle\psi_0|\psi_1\rangle| \le 2\sqrt{\delta(1-\delta)}$.*

**Fact 2.** *The class of degree-2 phase states $\left\{ \frac{1}{\sqrt{2^n}} \sum_x (-1)^{x^\top A x}|x\rangle \mid A \in \mathbb{F}_2^{n \times n} \right\}$ can be learned using $O(n)$ entangled measurements in time $O(n^3)$.*

*Proof.* The learning algorithm uses the Bell-sampling procedure: given two copies of $|\phi_A\rangle = \frac{1}{\sqrt{2^n}} \sum_x (-1)^{x^\top A x}|x\rangle$, perform $n$ CNOTs between the first copy and second copy and measure the second register to obtain a uniformly random $y \in \mathbb{F}_2^n$. The resulting quantum state can be written as

$$\frac{1}{\sqrt{2^n}} \sum_x (-1)^{x^\top A x + (x+y)^\top A(x+y)}|x\rangle = \frac{(-1)^{y^\top A y}}{\sqrt{2^n}} \sum_x (-1)^{x^\top (A+A^\top)\cdot y}|x\rangle.$$

The learning algorithm then applies the $n$-qubit Hadamard transform and measures to obtain bit string $(A + A^\top) \cdot y$. Repeating this process $O(n \log n)$ many times, one can learn $n$ linearly independent constraints about $A$. Using Gaussian elimination, this procedure allows the learner to learn the off-diagonal elements of $A$. In order to learn the diagonal elements of $A$ a learning algorithm applies the operation $|x\rangle \to (-1)^{x_i x_j}|x\rangle$ if $A_{ij} = 1$ for every $i \ne j$. The resulting quantum state is $\sum_x (-1)^{\sum_i x_i A_{ii}}|x\rangle$ and the learner can apply the $n$-qubit Hadamard transform to learn the diagonal elements of $A$. $\qquad\square$

**Fact 3.** *For distributions $p, q : \mathcal{X} \to [0,1]$, define $|\psi_p\rangle = \sum_{x \in \mathcal{X}} \sqrt{p(x)}|x\rangle$ and $|\psi_q\rangle$ similarly. Then*

$$d_{\mathsf{Tr}}(|\psi_p\rangle, |\psi_q\rangle)^2 \le 2d_{\mathsf{TV}}(p, q).$$

*Proof.* In order to see the fact, first note that we have $\langle\psi_p|\psi_q\rangle = \sum_x \sqrt{p(x)q(x)}$ and

$$d_{\mathsf{Tr}}(|\psi_p\rangle, |\psi_q\rangle)^2 = 1 - \langle\psi_p|\psi_q\rangle^2 = 1 - \left( \sum_x \sqrt{p(x)q(x)} \right)^2.$$

By the definition of the Hellinger distance, we have that $d_H(p,q)^2 = 1 - \sum_x \sqrt{p(x)q(x)}$, so we have

$$d_{\mathsf{Tr}}(|\psi_p\rangle, |\psi_q\rangle)^2 = 1 - \left(1 - d_H(p,q)^2\right)^2 = 2d_H(p,q)^2 - d_H(p,q)^4 \le 2d_H(p,q)^2 \le 2d_{\mathsf{TV}}(p,q),$$

where the final inequality used is [2, Proposition 1]. $\qquad\square$

**Fact 4** (Discriminating coherent encodings of distributions). *Consider $|\psi_p\rangle$ and $|\psi_q\rangle$ for some distributions $p$ and $q$ over $\mathcal{X}$ as defined above. Let $\phi$ be a function from $\mathcal{X}$ to $[-1, 1]$ We have:*

- $\max_{\phi: X \to [-1,1]} |\sum_x (p(x) - q(x))\phi(x)| = 2d_{\mathsf{TV}}(p, q)$.

*Similarly, let $|\psi\rangle$ and $|\phi\rangle$ be arbitrary quantum states and $M$ an obervable on their Hilbert space. We have:*

- $\max_{M, \|M\| \leq 1} |\mathsf{Tr}(M(\psi - \phi))| = 2d_{\mathsf{Tr}}(\psi, \phi)$.

*Proof.* 1. Choose $\phi = \delta(p(x) > q(x)) - \delta(q(x) \leq p(x))$, where $\delta(\cdot)$ is an indicator function from boolean clauses to $\{1, 0\}$ which evaluates to 1 if its argument evaluates to true and evaluates to 0 if its argument is false.

$$\left| \sum_{x \in X} (p(x) - q(x))\phi(x) \right| = \sum_{x \in X; p(x) > q(x)} (p(x) - q(x)) + \sum_{x \in X; p(x) \leq q(x)} (q(x) - p(x)) \tag{1}$$
$$= \sum_{x \in X} |p(x) - q(x)| = 2d_{\mathsf{TV}}(p, q).$$

2. See for example [3, Lemma 9.1.1]. The operator $\psi - \phi$ is Hermitian and can be diagonalized. Let $\Pi_+$ be the projector onto the positive part of its spectrum and $\Pi_-$ be the projector onto the negative part of its spectrum, so that $\Pi_- + \Pi_+ = \mathbb{I}$. Notice that:

$$2d_{Tr}(\psi, \phi) = \mathsf{Tr}(\Pi_+(\psi - \phi)) - \mathsf{Tr}(\Pi_-(\psi - \phi)) = 2\mathsf{Tr}(\Pi_+(\psi - \phi)). \tag{2}$$

$\square$

**Fact 5.** *For distinct $A, B \in \mathbb{F}_2^{n \times n}$, we have that $\Pr_{x \sim \{0,1\}^n}[x^\top A x \neq x^\top B x] \geq 1/4$.*

*Proof.* $\Pr_{x \sim \{0,1\}^n}[x^\top A x \neq x^\top B x] = \Pr_{x \sim \{0,1\}^n}[x^\top A x \oplus x^\top B x \neq 0] \geq \frac{1}{4}$, where the inequality follows from the Schwartz-Zippel lemma for Boolean functions [4]. $\square$

## 1.4 Learning models

In this section we first describe the learning models we will be concerned with in this paper.

**Classical** PAC **learning.** Valiant [5] introduced the classical Probably Approximately Correct (PAC) learning model. In this model, a *concept class* $\mathcal{C} \subseteq \{c : \{0,1\}^n \to \{0,1\}\}$ is a collection of Boolean functions. The learning algorithm $\mathcal{A}$ obtains *labelled examples* $(x, c(x))$ where $x \in \{0,1\}^n$ is uniformly random and $c \in \mathcal{C}$ is the *unknown* target function.[1] The goal of an $(\varepsilon, \delta)$-learning algorithm $\mathcal{A}$ is the following: for every $c \in \mathcal{C}$, given labelled examples $\{(x^i, c(x^i))\}$, with probability $\geq 1 - \delta$ (over the randomness of the labelled examples and the internal randomness of the algorithm), output a hypothesis $h : \{0,1\}^n \to \{0,1\}$ such that $\Pr_x[c(x) = h(x)] \geq 1 - \varepsilon$. The $(\varepsilon, \delta)$-sample complexity of a learning algorithm $\mathcal{A}$ is the maximal number of labelled examples used, maximized over all $c \in \mathcal{C}$. The $(\varepsilon, \delta)$-sample complexity of learning $\mathcal{C}$ is the minimal sample complexity over all $(\varepsilon, \delta)$-learners for $\mathcal{C}$. Similarly the $(\varepsilon, \delta)$-time complexity of learning $\mathcal{C}$ is the total number of time steps used by an optimal $(\varepsilon, \delta)$-learner for $\mathcal{C}$.

**Quantum** PAC **learning.** The quantum PAC (QPAC) model was introduced by Bshouty and Jackson [6] wherein, they allowed the learner access to quantum examples of the form

$$|\psi_c\rangle = \frac{1}{\sqrt{2^n}} \sum_{x \in \{0,1\}^n} |x, c(x)\rangle.$$

Note that measuring $|\psi_c\rangle$ in the computational basis produces a classical labelled example, so quantum examples are at least stronger then classical examples. Understanding their strength and

---

[1]More generally in PAC learning, there is an unknown distribution $D : \{0,1\}^n \to [0,1]$ from which $x$ is drawn. Throughout this paper we will be concerned with uniform-distribution PAC learning, i.e., $D$ is the uniform distribution, so we describe the learning model for the uniform distribution for simplicity.

weakness has been looked at by several works (we refer an interested reader to the survey [7]). Like the classical complexities, one can similarly define the $(\varepsilon, \delta)$-sample and time complexity for learning $\mathcal{C}$ as the quantum sample complexity (i.e., number of quantum examples $|\psi_c\rangle$) used and quantum time complexity (i.e., number of quantum gates used in the algorithm) of an optimal $(\varepsilon, \delta)$-learner for $\mathcal{C}$.

**Quantum** PAC **learning with classification noise.** Classically, the $\eta$-classification noise model is defined as follows: for an unknown $c \in \mathcal{C}$, a learning algorithm is given uniformly random $x \in \{0,1\}^n$ and $b \in \{0,1\}$ where, $b = c(x)$ with probability $1 - \eta$ and $b = \overline{c(x)}$ with probability $\eta$. In the same work, Bshouty and Jackson [6] defined quantum learning with classification noise, wherein a learning algorithm is given access to

$$|\psi_c^\eta\rangle = \frac{1}{\sqrt{2^n}} \sum_{x \in \{0,1\}^n} |x\rangle \otimes (\sqrt{1-\eta}|c(x)\rangle + \sqrt{\eta}|\overline{c(x)}\rangle).$$

Such quantum examples have been investigated in prior works [6, 8, 9].

**Learning with entangled and separable measurements.** Observe that in the usual definition of QPAC above, a learning algorithm is given access to $|\psi_c\rangle^{\otimes T}$ and needs to learn the unknown $c \in \mathcal{C}$. In this paper we make the distinction between the case where the learner uses entangled measurements, i.e., perform an arbitrary operation on copies of $|\psi_c\rangle$ versus the setting where the learner uses separable measurements, i.e., performs a single-copy measurement on every copy of $|\psi_c\rangle$ in the learning algorithm. When discussing learning with entangled and separable measurements, in this paper we will be concerned with *exact* learning, i.e., with probability $\geq 2/3$, the learner needs to *identify c*. We denote EntExact as the sample complexity of learning with entangled measurements and SepExact as the sample complexity of learning with separable measurements.

**Quantum statistical query learning.** We now discuss the QSQ model, following the definitions given in [10]. We first discuss the classical statistical query (SQ) model for learning an unknown concept $c \in \mathcal{C}$. Classically, the learner has access to a *statistical query oracle* Stat, that on input a function $\phi : \{0,1\}^{n+1} \to [-1,1]$ and a *tolerance* $\tau$ and returns a number $\alpha$ satisfying

$$\left| \alpha - \mathop{\mathbb{E}}_{x \sim \{0,1\}^n} [\phi(x, c(x))] \right| \leq \tau .$$

A classical SQ algorithm can adaptively make a sequence of Stat queries $\{(\phi_i, \tau_i)\}_i$ and based on the responses $\{\alpha_i\}$ outputs a hypothesis $h : \{0,1\}^n \to \{0,1\}$. The goal of the classical SQ algorithm is to, with probability at least $1 - \delta$, output an $h$ such that $\Pr_x[h(x) = c(x)] \geq 1 - \varepsilon$. The query complexity of a classical SQ algorithm is the number of Stat queries the algorithm makes and the time complexity is the total number of gates used by the algorithm and in the description of the hypothesis.

A natural way to extend the learning model is to allow the algorithm *quantum statistical queries*. In the classical case, one can think of the input $\phi$ to the Stat oracle as a specification of a *statistic* about the distribution of examples $(x, c(x))$, and the output of the Stat oracle is an *estimation* of $\phi$: one can imagine that the oracle receives i.i.d. labeled examples $(x, c(x))$ and empirically computes an estimate of $\phi$, which is then forwarded to the learning algorithm. In the quantum setting, one can imagine the analogous situation where the oracle receives copies of the quantum example state $|\psi_c\rangle$, and performs a measurement indicated by the *observable* $M$ on each copy and outputs an estimate of $\langle \psi_c | M | \psi_c \rangle$.

Generally, in order to learn an unknown quantum state $\rho$ in the QSQ model the learner makes Qstat queries taking as input an observable $M$ and tolerance $\tau$ and recieves a $\tau$-approximation of $\text{Tr}[M\rho]$, i.e.,

$$\text{Qstat} : (M, \tau) \mapsto \alpha \in [\text{Tr}(M\rho) + \tau, \text{Tr}(M\rho) - \tau].$$

In the case of quantum example states, $\rho = |\psi_c\rangle\langle\psi_c|$, $M$ is an operator on $\mathbb{C}^{2^{n+1}}$, and the Qstat oracle functions as

$$\text{Qstat} : (M, \tau) \mapsto \alpha \in [\langle\psi_c|M|\psi_c\rangle + \tau, \langle\psi_c|M|\psi_c\rangle - \tau].$$

Like in the classical SQ model, the goal of the QSQ learner is still to output a hypothesis $h : \{0,1\}^n \to \{0,1\}$ that satisfies $\Pr_{x \sim \{0,1\}^n}[h(x) = c(x)] \geq 1 - \varepsilon^2$. We emphasize that the learning algorithm is still a *classical* randomized algorithm and only receives statistical estimates of measurements on quantum examples. The quantum query complexity of the QSQ algorithm is the number of Qstat queries the algorithm makes and the quantum time complexity is the total number of gates used by the algorithm and in the description of the hypothesis. There are three ways to motivate the QSQ model

1. Clearly any binary measurement $\{M, I - M\}$ can be simulated with a Qstat query to $M$ or $I - M$. In the opposite direction, any observable $M$ such that $\|M\| \leq 1$ can be converted into the POVM $\left\{ \frac{I+M}{2}, \frac{I-M}{2} \right\}$. Thus, Qstat queries as defined above and binary POVMs are essentially the same model. From a theoretical perspective, performing 2-outcome measurements are weaker (and easier to implement) than arbitrary separable measurements, which is in turn weaker (and easier to implement) than entangled measurements, so it is useful to understand the power of expectation values in quantum learning theory and the QSQ captures this question in a theoretical framework.

2. One could envision a situation where quantum states $\rho$ are prepared in the "cloud" and the *classical* learning algorithm needs to only interact with the cloud *classically*. An efficient QSQ model allows a quantum advantage in learning in this framework.

3. The QSQ model naturally extends recent works [11–13] wherein they consider the limitations of classical SQ algorithms for learning a quantum state $\psi_U = U|0^n\rangle$, i.e. they consider the model where $M$ is diagonal specifiable as $M = \sum_x \phi(x)|x\rangle\langle x|$, then

$$\langle \psi_U | M | \psi_U \rangle = \sum_x \phi(x)\langle x|U|0^n\rangle^2 = \sum_x \phi(x)P_U(x) = \mathbb{E}_{x \sim P_U}[\phi(x)],$$

which is precisely $\alpha_\phi$ they assume access to, in order to learn the unknown $U$

*Throughout this paper*, for notational convenience we use the following notation: $(i)$ for an $n$-bit problem, when we do not specify a tolerance for the Qstat oracle, we implicitly assume that the tolerance is $\tau = 1/\operatorname{poly}(n)$, $(ii)$ we always make Qstat queries with an operator $M$ that satisfies $\|M\| \leq 1$, so we do not explicitly state this when discussing Qstat queries, $(iii)$ We say a $n$-bit concept class $\mathcal{C}$ is QSQ learnable if $\mathcal{C}$ can be learned using $\operatorname{poly}(n)$ many Qstat queries, each with tolerance $\tau = 1/\operatorname{poly}(n)$ and observable $M$ which is implementable using $\operatorname{poly}(n)$ many gates.

## 2 Relating separable and entangled measurements

Before proving our main theorem, we will use the following proposition, which was proven earlier in [14] in the context of learning quantum channels.

**Proposition 1.** *Let $\mathcal{C} \subseteq \{c : \{0,1\}^n \to \{0,1\}\}$ and $\varepsilon > 0$. Given*

$$T = O\left( \frac{\log |\mathcal{C}| + \log 1/\delta}{\varepsilon} \right) \tag{3}$$

*copies of $|\psi_c\rangle = \frac{1}{\sqrt{2^n}} \sum_x |x, c(x)\rangle$ for an unknown $c \in \mathcal{C}$, there exists an algorithm that uses separable measurements and with probability $\geq 1 - \delta$, outputs a $c' \in \mathcal{C}$ such that $\Pr_{x \sim \{0,1\}^n}[c(x) = c'(x)] \geq 1 - \varepsilon$.*

Note that their proposition deals with general states and $T = O\left( \frac{\log |\mathcal{C}| + \log 1/\delta}{\varepsilon^2} \right)$. For our purposes the $\frac{1}{\varepsilon^2}$ is improved to $\frac{1}{\varepsilon}$ by the fact that $d_{\mathsf{Tr}}(\rho_f, \rho_h) = \varepsilon$ implies that $\Pr_{x \sim \{0,1\}^n}[f(x) \neq h(x)] = \Theta(\sqrt{\varepsilon})$.

**Theorem 3.** *Let $\mathcal{C}$ be a concept class $\mathcal{C} \subseteq \{c : \{0,1\}^n \to \{0,1\}\}$ and*

$$\eta_m = \min_{c,c' \in \mathcal{C}} \Pr_{x \sim \{0,1\}}[c(x) \neq c'(x)], \quad \eta_a = \mathbb{E}_{c,c' \in \mathcal{C}} \Pr_{x \sim \{0,1\}}[c(x) \neq c'(x)].$$

---

[2]Generally we relax this assumption and implicitly assume that the learner is outputting a quantum state that is close in trace distance to the unknown state. This is a more general model and clearly lower bounds then carry over to algorithms that output classical functions as well.

*Then we have that*

$$\mathsf{SepExact}(\mathcal{C}) \leq O\Big(n \cdot \mathsf{EntExact}(\mathcal{C}) \cdot \min\big\{\eta_a/\eta_m \, , \, \mathsf{EntExact}(\mathcal{C})\big\}\Big).$$

*Furthermore, there exists $\mathcal{C}$ for which this inequality is tight.*

*Proof.* First observe that

$$\mathsf{SepExact} \leq 2/\eta_m \cdot \log|\mathcal{C}|. \tag{4}$$

This is easy to see: fix $\varepsilon = \eta_m/2$ in Proposition 1 and consider a separable approximate algorithm that, given copies of $|\psi_c\rangle$, an approximate learning algorithm outputs $c'$ such that $\Pr_{x\sim\{0,1\}^n}[c(x) \neq c'(x)] \leq \varepsilon$, then $c = c'$ by definition of $\eta_m$, hence this algorithm is a separable *exact* learning algorithm.

Next, we prove two lower bounds on $\mathsf{EntExact}$:

$$\mathsf{EntExact} \geq \max\left\{\frac{1}{\eta_m}, \frac{\log|\mathcal{C}|}{n\eta_a}\right\} \tag{5}$$

To see the first lower bound in Eq. (5), observe the following: consider the $c, c' \in \mathcal{C}$ for which $\Pr_{x\sim\{0,1\}}[c(x) \neq c'(x)] = \eta_m$, then every exact learning algorithm needs to distinguish between $c, c'$. Since $\langle\psi_c|\psi_{c'}\rangle = 1 - \eta_m$, by Fact 1, this implies a lower bound of $T = \Omega(1/\eta_m)$ many quantum examples to distinguish between $c, c'$ with bias $\Omega(1)$.

To see the second lower bound in Eq. (5), first note that $1 - \eta_a = \mathbb{E}_{c,c'\in\mathcal{C}}\mathbb{E}_{x\sim\{0,1\}}[c(x) = c'(x)]$. Next, observe that

$$\mathsf{EntExact} \geq \frac{\log|\mathcal{C}|}{n\eta_a}. \tag{6}$$

The proof of this is similar to the information-theoretic proof in [8]. We prove the lower bound for $\mathcal{C}$ using a three-step information-theoretic technique. Let $\mathbf{A}$ be a random variable that is uniformly distributed over $\mathcal{C}$. Suppose $\mathbf{A} = c_V$, and let $\mathbf{B} = \mathbf{B}_1 \ldots \mathbf{B}_T$ be $T$ copies of the quantum example

$$|\psi_c\rangle = \frac{1}{\sqrt{2^n}} \sum_{x\in\{0,1\}^n} |x, c(x)\rangle$$

for $c \in \mathcal{C}$. The random variable $\mathbf{B}$ is a function of the random variable $\mathbf{A}$. The following upper and lower bounds on $I(\mathbf{A} : \mathbf{B})$ are similar to [8, Theorem 12] and we omit the details of the first two steps here.

1. $I(\mathbf{A} : \mathbf{B}) \geq \Omega(\log|\mathcal{C}|)$ because $\mathbf{B}$ allows one to recover $\mathbf{A}$ with high probability.

2. $I(\mathbf{A} : \mathbf{B}) \leq T \cdot I(\mathbf{A} : \mathbf{B}_1)$ using a chain rule for mutual information.

3. $I(\mathbf{A} : \mathbf{B}_1) \leq O(n \cdot \eta_a)$.
   *Proof (of 3).* Since $\mathbf{A}\mathbf{B}$ is a classical-quantum state, we have

   $$I(\mathbf{A} : \mathbf{B}_1) = S(\mathbf{A}) + S(\mathbf{B}_1) - S(\mathbf{A}\mathbf{B}_1) = S(\mathbf{B}_1),$$

   where the first equality is by definition and the second equality uses $S(\mathbf{A}) = \log|\mathcal{C}|$ since $\mathbf{A}$ is uniformly distributed over $\mathcal{C}$, and $S(\mathbf{A}\mathbf{B}_1) = \log|\mathcal{C}|$ since the matrix

   $$\sigma = \frac{1}{|\mathcal{C}|} \sum_{c\in\mathcal{C}} |c\rangle\langle c| \otimes |\psi_c\rangle\langle\psi_c|$$

   is block-diagonal with $|\mathcal{C}|$ rank-1 blocks on the diagonal. It thus suffices to bound the entropy of the (vector of singular values of the) reduced state of $\mathbf{B}_1$, which is

   $$\rho = \frac{1}{|\mathcal{C}|} \sum_{c\in\mathcal{C}} |\psi_c\rangle\langle\psi_c|.$$

Let $\sigma_0 \geq \sigma_1 \geq \cdots \geq \sigma_{2^{n+1}-1} \geq 0$ be the singular values of $\rho$. Since $\rho$ is a density matrix, these form a probability distribution. Now observe that $\sigma_0 \geq 1 - \eta_a$: consider the vector $u = \frac{1}{|\mathcal{C}|} \sum_{c' \in \mathcal{C}} |\psi_{c'}\rangle$ and observe that

$$
\begin{aligned}
u^\top \rho u &= \frac{1}{|\mathcal{C}|^3} \sum_{c,c',c'' \in \mathcal{C}} \langle \psi_c | \psi_{c'} \rangle \langle \psi_c | \psi_{c''} \rangle \\
&= \mathbb{E}_c \Big[ \mathbb{E}_{c'} [\langle \psi_c | \psi_{c'} \rangle] \Big] \cdot \Big[ \mathbb{E}_{c''} [\langle \psi_c | \psi_{c''} \rangle] \Big] \\
&\geq \Big( \mathop{\mathbb{E}}_{c,c'} [\langle \psi_c | \psi_{c'} \rangle] \Big) \cdot \Big( \mathop{\mathbb{E}}_{c,c''} [\langle \psi_c | \psi_{c''} \rangle] \Big) = \Big( \mathop{\mathbb{E}}_{c,c' \in \mathcal{C}} \mathop{\Pr}_{x \sim \{0,1\}} [c(x) = c'(x)] \Big)^2 \geq 1 - 2\eta_a,
\end{aligned}
$$

where the first inequality is by Chebyshev's sum inequality (since all the inner products are non-negative) and the second inequality follows from the definition of $\eta_a$. Hence we have that $\sigma_0 = \max_u \{ u^\top \rho u / u^\top u \} \geq 1 - 2\eta_a$ (where we used that $\|u\|_2 \leq 1$). Let $\mathbf{N} \in \{0, 1, \ldots, 2^{n+1} - 1\}$ be a random variable with probabilities $\sigma_0, \sigma_1, \ldots, \sigma_{2^{n+1}-1}$, and $\mathbf{Z}$ an indicator for the event "$\mathbf{N} \neq 0$." Note that $\mathbf{Z} = 0$ with probability $\sigma_0 \geq 1 - 2\eta_a$, and $H(\mathbf{N} \mid \mathbf{Z} = 0) = 0$. By a similar argument as in [8, Theorem 15], we have

$$
\begin{aligned}
S(\rho) &= H(\mathbf{N}) = H(\mathbf{N}, \mathbf{Z}) = H(\mathbf{Z}) + H(\mathbf{N} \mid \mathbf{Z}) \\
&= H(\sigma_0) + \sigma_0 \cdot H(\mathbf{N} \mid \mathbf{Z} = 0) + (1 - \sigma_0) \cdot H(\mathbf{N} \mid \mathbf{Z} = 1) \\
&\leq H(\eta_a) + \eta_a(n + 1) \\
&\leq O(\eta_a(n + \log(1/\eta_a)))
\end{aligned}
$$

using $H(\alpha) \leq O(\alpha \log(1/\alpha))$.

Combining these three steps implies $T = \Omega(\log |\mathcal{C}| / (n\eta_a))$.

Now putting the relations between $\mathsf{EntExact}, \mathsf{SepExact}$ together we get

$$
\mathsf{SepExact} \leq n \cdot \eta_a / \eta_m \cdot \mathsf{EntExact} \leq n \cdot \mathsf{EntExact}^2,
$$

hence we have the desired upper bound as in the theorem statement[3]

$$
\mathsf{SepExact} \leq O\Big( n \cdot \mathsf{EntExact} \cdot \min \big\{ \eta_a / \eta_m, \mathsf{EntExact} \big\} \Big).
$$

To show that this inequality is optimal, observe that: if $\mathcal{C}$ is the class of degree-2 phase states, i.e., $\mathcal{C} = \{ f_A(x) = x^\top A x : A \in \{0,1\}^{n \times n} \}$, then $\eta_m = \eta_a = \Theta(1)$ by Fact 5. We saw in Fact 2 that this class can be learned using $\Theta(n)$ entangled measurements, so $\mathsf{EntExact} = \Theta(n)$ and the above upper bound implies $\mathsf{SepExact} = O(n^2)$, which was shown to be optimal in [15]. □

## 3 Lower bounds for Quantum statistical query learning

Here we prove our main theorem which provides combinatorial quantities one can use to lower bound the QSQ complexity of learning. The techniques and parameters used in this section are inspired by several seminal classical works on classical SQ learning [16–19]. We first define *statistical decision problems*, then define the quantum statistical dimension which lower bounds the decision problem complexity and finally discuss the variance and average correlation bound to lower bound the QSD dimension. For notational convenience we discuss some notation we use in this section: throughout this section, we will let $\mathcal{C}$ be a collection of $n$-qubit quantum states. We let $\mathsf{QSQ}(\mathcal{C})$ be the complexity of learning $\mathcal{C}$ using Qstat queries, QSD is the complexity of the *decision* problem, QQC is the quantum statistical dimension, QAC is the average correlation bound.

### 3.1 Learning is as hard as deciding

**Definition 1** (Quantum (many-vs-one) Decision Problem). *Let $\tau \in [0, 1]$ and let $\sigma \notin \mathcal{C}$. A quantum statistical decision problem for $(\mathcal{C}, \sigma)$ is defined as: for an unknown state $\rho$, given $\mathsf{Qstat}(\tau)$ access to $\rho$ decide if $\rho \in \mathcal{C}$ or $\rho = \sigma$. Let $\mathsf{QQC}_\tau(\mathcal{C}, \sigma)$ be the number of $\mathsf{Qstat}(\tau)$ queries made by the best algorithm for the decision problem that succeeds with probability at least $1 - \delta$.*

---

[3]We state the theorem as below, since it is apriori unclear as to why $1/\eta_m$ is a lower bound on $\mathsf{EntExact}$.

We now prove our first lemma that QQC is actually a lower bound on QSQ learning the concept class $\mathcal{C}$. We remark that a similar lemma appears for *classical* SQ in [13], we want to thank the authors for sharing their manuscript during the completion of our work.

**Lemma 1** (Learning is at least as hard as deciding). *Let $\varepsilon \geq \tau > 0$ and $\sigma \notin \mathcal{C}$ be such that $\min_{\rho \in \mathcal{C}}[d_{\mathsf{Tr}}(\rho, \sigma)] > 2(\tau + \varepsilon)$. Let $\mathsf{QSQ}_\tau^{\varepsilon,\delta}(\mathcal{C})$ be the number of $\mathsf{Qstat}(\tau)$ queries made by a QSQ algorithm that on input $\rho$ outputs $\pi$, such that $d_{\mathsf{Tr}}(\pi, \rho) \leq \varepsilon$ with probability $\geq 1 - \delta$. Then*

$$\mathsf{QSQ}_\tau^{\varepsilon,\delta}(\mathcal{C}) \geq \mathsf{QQC}_\tau(\mathcal{C}, \sigma) - 1.$$

*Proof.* We show this by solving the statistical quantum decision problem by querying a $\mathsf{QSQ}_\tau^{\varepsilon,\delta}(\mathcal{C})$ learning algorithm $\mathcal{A}$. For $\rho \in \mathcal{C}$, $\mathcal{A}$ outputs, with probability $\geq 1 - \delta$, a classical description of quantum state $\pi$ such that $d_{\mathsf{Tr}}(\rho, \pi) \leq \varepsilon$. Note that for $\sigma \notin \mathcal{C}$, the output of $\mathcal{A}$ is not well-defined and we assume that $\mathcal{A}$ can output anything.

Let the output of $\mathcal{A}$ be $\pi$. If $\pi$ is not a valid quantum state, return "$\rho = \sigma$". We then check if $\min_{\rho \in \mathcal{C}} d_{\mathsf{Tr}}(\pi, \rho) > \varepsilon$. If yes, return "$\rho = \sigma$". At this point, we know the classical description of both $\pi$ and $\sigma$ and also know that there exists some $\nu \in \mathcal{C}$, such that $d_{\mathsf{Tr}}(\pi, \nu) \leq \varepsilon$. We can find such $\nu$ that is closest to $\pi$, as well as an operator $\Pi_+$ which is a projector onto the positive part of the spectrum of the hermitian operator $\nu - \sigma$. Finding this may be computationally difficult, but does not require additional $\mathsf{Qstat}(\tau)$ queries. We then query $\mathsf{Qstat}(\tau)$ with $\Pi_+$ to obtain a response $R$. If $|R - \mathsf{Tr}(\Pi_+\sigma)| \leq \tau$, return "$\rho = \sigma$". Return "$\rho \in \mathcal{C}$" otherwise.

The algorithm outputs "$\rho = \sigma$" on all inputs $\rho = \sigma$ with certainty. On input $\rho \in \mathcal{C}$, the algorithm $\mathcal{A}$ returns, with probability at least $(1 - \delta)$ a description of a state $\pi$ that is $\varepsilon$ close to the input. Our algorithm then uses this information to find a state $\nu \in \mathcal{C}$, such that $d_{\mathsf{Tr}}(\pi, \nu) \leq \varepsilon$. We have from reverse triangle inequality that:

$$|\mathsf{Tr}(\Pi_+(\rho - \sigma))| \geq |\underbrace{|\mathsf{Tr}(\Pi_+(\sigma - \nu))|}_{d_{\mathsf{Tr}}(\sigma,\nu)} - \underbrace{|\mathsf{Tr}(\Pi_+(\nu - \rho))|}_{<2\varepsilon}| \geq d_{\mathsf{Tr}}(\nu, \sigma) - 2\varepsilon > 2\tau, \qquad (7)$$

where we used that $|\mathsf{Tr}(\Pi_+(\nu - \rho))| \leq d_{\mathsf{Tr}}(\nu, \rho) \leq d_{\mathsf{Tr}}(\nu, \pi) + d_{\mathsf{Tr}}(\rho, \pi) < 2\varepsilon$.[4] It follows that:

$$|R - \mathsf{Tr}(\Pi_+\sigma)| \geq |\mathsf{Tr}(\Pi_+(\rho - \sigma))| - \underbrace{|R - \mathsf{Tr}(\Pi_+\rho)|}_{\leq \tau} > \tau, \qquad (8)$$

The algorithm outputs "$\rho \in \mathcal{C}$" with probability at least $(1 - \delta)$, as expected. $\qquad\square$

For completeness, we also include a proof of a lower bound on the learning complexity by a decision problem hidden completely (that is, $\sigma \in \mathcal{C}$) inside of $\mathcal{C}$.

**Lemma 2** (Learning is as hard as deciding, alternative take). *Let $\mathcal{D} \subset \mathcal{C}$ and let $\sigma \in \mathcal{C}$, $\sigma \notin \mathcal{D}$, $d_{\mathsf{Tr}}(\mathcal{D}, \sigma) > \varepsilon$ and $\varepsilon \geq \tau > 0$. Then:*

$$\mathsf{QSQ}_\tau^{\varepsilon,\delta}(\mathcal{C}) \geq \mathsf{QQC}_\tau(\mathcal{D}, \sigma). \qquad (9)$$

*Proof.* Let $\mathcal{A}$ be a statistical $\varepsilon, \delta$ learning algorithm for $\mathcal{C}$ that uses $\mathsf{Qstat}(\tau)$ queries. On input $\rho \in \mathcal{C}$, the algorithm $\mathcal{A}$ outputs (with probability at least $1 - \delta$) a state $\nu$, such that $d_{\mathsf{Tr}}(\nu, \rho) \leq \varepsilon$ and uses $\mathsf{QSQ}_\tau^{\varepsilon,\delta}(\mathcal{C})$ many queries. Output "$\rho = \sigma$" if $d_{\mathsf{Tr}}(\nu, \sigma) < \varepsilon$, otherwise output "$\rho \in \mathcal{D}$". The algorithm clearly succeeds with probability at least $1 - \delta$. $\qquad\square$

## 3.2 Quantum statistical dimension to bound the decision problem

With this lemma, in order to lower bound QSQ learning it suffices to lower bound QQC, which we do via the quantum statistical dimension that we define now.

**Definition 2** (Quantum Statistical Dimension). *Let $\tau \in [0, 1]$ and $\mu$ be a distribution over a set of $n$-qubit quantum states $\mathcal{C}$ and $\sigma \notin \mathcal{C}$ be an $n$-qubit state. Define the maximum covered fraction:*

$$\kappa_\tau\text{-}frac(\mu, \sigma) = \max_{M:\|M\|\leq 1}\left\{ \Pr_{\rho\sim\mu}\left[|\mathsf{Tr}(M(\rho - \sigma))| > \tau\right]\right\}. \qquad (10)$$

---

[4]Note that this inequality is maximized if $\nu \neq \rho$. This can happen if the input state $\rho \in \mathcal{C}$ is less than $2\varepsilon$ far from another state $\rho' \in \mathcal{C}$ and the learning algorithm outputs $\pi$ that is closer to $\rho' \in \mathcal{C}$.

*The quantum statistical dimension is:*

$$\mathsf{QSD}_\tau(\mathcal{C}, \sigma)) = \sup_\mu \left[\kappa_\tau \text{-}\textit{frac}(\mu, \sigma)\right]^{-1}, \tag{11}$$

*where the supremum is over distibutions over $\mathcal{C}$.*

This definition is essentially the same as Feldman's definition of randomized statistical dimension in [18], but uses the difference between the expectation values of quantum observables. Sections 4, 5 and 6 of our work show that this has many nontrivial and interesting consquences. The following lemma, following similarly from Feldman's work [18, Lemma 3.8], will be convenient later:

**Lemma 3.** *Let $\tau > 0$, $\mathcal{C}$ be a set of quantum states and $\sigma \notin \mathcal{C}$ be another quantum state. Let $d$ be the smallest integer such that: there exists a distribution $\nu$ over* Qstat *queries $M$ satisfying*

$$\forall\, \rho \in \mathcal{C} : \quad \Pr_{M \sim \nu} \left[|\mathsf{Tr}(M(\rho - \sigma))| > \tau\right] \geq 1/d,$$

*then $d = \mathsf{QSD}_\tau(\mathcal{C}, \sigma)$.*

*Proof.* See also [18, Lemma 3.8.]. Suppose that the Qstat tolerance is fixed to $\tau$ and let $\mathcal{M}$ be the set of all valid Qstat queries. Define $G : \mathcal{C} \times \mathcal{M} \to \{0, 1\}$ as $G(\rho, M) = \delta[|\mathsf{Tr}(M(\rho - \sigma))| > \tau]$, where $\delta[\cdot]$ is the indicator function. Let $\mu$ be distribution over $\mathcal{C}$ and let $\nu$ be distribution over $\mathcal{M}$. Consider the bilinear function:

$$F(\mu, \nu) = \int_{\mathcal{M}} d\nu(M) \sum_{\rho \in \mathcal{C}} \mu(\rho) G(M, \rho) = \Pr_{\rho \sim \mu} \Pr_{M \sim \nu} [|\mathsf{Tr}(M(\rho - \sigma))| > \tau]. \tag{12}$$

It follows by von Neumann's minimax theorem that:

$$\min_\mu \max_\nu F(\mu, \nu) = \max_\nu \min_\mu F(\mu, \nu) =: 1/d, \tag{13}$$

where the optimization is over possible distributions $\mu$ over $\mathcal{C}$ and distributions $\nu$ over $\mathcal{M}$. For a distribution $\mu$ over $\mathcal{C}$, there exists an optimal distinguishing measurement $M \in \mathcal{M}$, from which:

$$\min_\mu \max_\nu F(\mu, \nu) = \min_\mu \max_{M \in \mathcal{M}} \Pr_{\rho \sim \mu}[|\mathsf{Tr}(M(\rho - \sigma))| > \tau]. \tag{14}$$

Observe that:

$$d = \sup_\mu (\max_{M \in \mathcal{M}} \Pr_{\rho \sim \mu}[|\mathsf{Tr}(M(\rho - \sigma))| > \tau])^{-1}, \tag{15}$$

which is the definition of $\mathsf{QSD}_\tau$ by Eq. 11. Similarly, we have that:

$$d = (\max_\nu \min_\mu F(\mu, \nu))^{-1} = \inf_\nu (\min_{\rho \in \mathcal{C}} \Pr_{\rho \sim \mu}[|\mathsf{Tr}(M(\rho - \sigma))| > \tau])^{-1}. \tag{16}$$

For a given distribution $\nu$ over $\mathcal{M}$, it then holds for all $\rho \in \mathcal{C}$ that $\Pr_{M \sim \nu}[|\mathsf{Tr}(M(\rho - \sigma))| > \tau] \geq 1/d$. This is the definition in Lemma 3. $\qquad\square$

We now show that the QQC complexity is lower bounded by QSD.

**Lemma 4.** *For every $\sigma \notin \mathcal{C}$ and $\tau \in [0, 1]$, we have that*

$$\mathsf{QQC}_\tau(\mathcal{C}, \sigma) \geq (1 - 2\delta)\mathsf{QSD}_\tau(\mathcal{C}, \sigma). \tag{17}$$

*Proof.* Let $\mathcal{A}$ be the best algorithm that solves $(\mathcal{C}, \sigma)$ with probability at least $1 - \delta$ using $q$ Qstat$(\tau)$ queries $M_1, \ldots M_q$ chosen according to the internal randomness of $\mathcal{A}$. Suppose, for contradiction, that the response to every such query was $\mathsf{Tr}(M_i \sigma)$. Let $p_\rho = \Pr_{\mathcal{A}}[\exists i \in [q]||\mathsf{Tr}(M_i(\rho - \sigma))| > \tau]$ be the probability that $\rho \in \mathcal{C}$ can be distinguished from $\sigma$ by at least one of queries. If $p_\rho \leq 1 - 2\delta$, then with probability $2\delta$, the responses $\mathsf{Tr}(M_i \sigma)$ are valid Qstat$(\tau)$ responses. By correctness, on input $\rho = \sigma$, the algorithm can output "$\rho \in \mathcal{C}$" with probability at most $\delta$. This however means that on input $\rho \in \mathcal{C}$, the algorithm can output "$\rho \in \mathcal{C}$" with probability at most $\delta$ (since the responses did not change), which contradicts the algorithm correctness. It follows that $p_\rho \geq 1 - 2\delta$ for every $\rho \in \mathcal{C}$ and with probability $1 - 2\delta$, there exists a $M_i$ that distinguishes $\rho$ and $\sigma$. Running $\mathcal{A}$ and picking one of its queries uniformly randomly then gives $\Pr_M[|\mathsf{Tr}(M(\rho - \sigma)| > \tau] \geq \frac{1 - 2\delta}{q}$. Lemma 3 then implies that $q \geq (1 - 2\delta)\mathsf{QSD}_\tau(\mathcal{C}, \sigma)$. $\qquad\square$

## 3.3 Variance and average correlation lower bound quantum statistical dimension

We now present our main lower bound theorem, wherein we show that there are two combinatorial parameters that can be used to lower bound QSD, which in turn lower bounds sample complexity in the QSQ model. Throughout the paper we will use these two two parameters to prove our QSQ lower bounds.

**Theorem 4** (Lower bounds). *Let $\tau > 0$, $\mathcal{C}$ be the class of $n$-qubit states. Then*

1. *Variance bound: Let $\mu$ be a distribution over $\mathcal{C}$, such that $\mathbb{E}_{\rho \sim \mu}[\rho] \notin \mathcal{C}$. Then:*

$$\mathsf{QSD}_\tau(\mathcal{C}, \mathbb{E}_{\rho \sim \mu}[\rho]) \geq \tau^2 \cdot \min_{M, \|M\| \leq 1} \left( \mathsf{Var}_{\rho \sim \mu}[\mathsf{Tr}[\rho M]] \right)^{-1}, \tag{18}$$

*where*

$$\mathsf{Var}_{\rho \sim \mu}[\mathsf{Tr}(\rho M)] = \mathbb{E}_{\rho \sim \mu}[\mathsf{Tr}(\rho M)^2] - \left( \mathbb{E}_{\rho \sim \mu}[\mathsf{Tr}(\rho M)] \right)^2.$$

2. *Average correlation: For a full-rank quantum state $\sigma \notin \mathcal{C}$, define $\hat{\rho} := (\rho \sigma^{-1} - \mathbb{I})$ and:*

$$\gamma(\mathcal{C}, \sigma) = \frac{1}{|\mathcal{C}|^2} \sum_{\rho_1, \rho_2 \in \mathcal{C}} |\mathsf{Tr}(\hat{\rho}_1 \hat{\rho}_2 \sigma)|, \quad \kappa_\tau^\gamma\text{-}frac(\mathcal{C}_0, \sigma) := \max_{\mathcal{C}' \subseteq \mathcal{C}_0} \left\{ \frac{|\mathcal{C}'|}{|\mathcal{C}_0|} : \gamma(\mathcal{C}', \sigma) > \tau \right\}. \tag{19}$$

*Let $\mathsf{QAC}_\tau(\mathcal{C}, \sigma) = \sup_{\mathcal{C}_0 \subseteq \mathcal{C}} (\kappa_\tau^\gamma\text{-}frac(\mathcal{C}_0, \sigma))^{-1}$. Then, $\mathsf{QSD}_\tau(\mathcal{C}, \sigma) \geq \mathsf{QAC}_{\tau^2}(\mathcal{C}, \sigma)$.*

*Proof.* 1. We first prove the Let $\mu$ be a distribution over $\mathcal{C}$ and $M$ be a hermitian operator such that $\|M\| \leq 1$. By Chebyshev's inequality, we have that:

$$\Pr_{\rho \sim \mu} \left[ |\mathsf{Tr}(\rho M) - \mathbb{E}_{\rho \sim \mu}[\mathsf{Tr}(\rho M)]| \geq \tau \right] \leq \mathsf{Var}_{\rho \sim \mu}[\mathsf{Tr}(\rho M)] \cdot \tau^{-2}, \tag{20}$$

where

$$\mathsf{Var}_{\rho \sim \mu}[\mathsf{Tr}(\rho M)] = \mathbb{E}_{\rho \sim \mu}[\mathsf{Tr}(\rho M)^2] - \left( \mathbb{E}_{\rho \sim \mu}[\mathsf{Tr}(\rho M)] \right)^2.$$

Let $\nu$ be a distribution over the queries $M$ that are made by a randomized algorithm for the many-one distinguishing problem $(\mathcal{C}, \mathbb{E}_{\rho \sim \mathcal{C}}[\rho])$. Lemma 3 then implies for $d = \mathsf{QSD}_\tau(\mathcal{C}, \mathbb{E}_{\rho \sim \mathcal{C}}[\rho])$ that:[5]

$$\frac{1}{d} \leq \Pr_{M \sim \nu} \Pr_{\rho \sim \mu} [|\mathsf{Tr}(\rho M) - \mathbb{E}_{\rho \sim \mu}[\mathsf{Tr}(\rho M)]| \geq \tau] \leq \Pr_{M \sim \nu} \frac{\mathsf{Var}_{\rho \sim \mu}[\mathsf{Tr}(\rho M)]}{\tau^2}. \tag{21}$$

Since this inequality holds for any such distribution $\nu$, it holds for every query $M$. Hence,

$$\mathsf{QSD}_\tau(\mathcal{C}, \sigma) \geq \min_{M, \|M\| \leq 1} \frac{\tau^2}{\mathsf{Var}_{\rho \sim \mu}[\mathsf{Tr}[\rho M]]}. \tag{22}$$

2. We now prove the average correlation bound. Let $\mathcal{C}'$ be a set of quantum states. Let $\hat{\rho} := (\rho \sigma^{-1} - \mathbb{I})$ and define:

$$\gamma(\mathcal{C}', \sigma) = \frac{1}{|\mathcal{C}'|^2} \sum_{\rho_i, \rho_j \in \mathcal{C}'} |\mathsf{Tr}[\hat{\rho}_i \hat{\rho}_j \sigma]| \tag{23}$$

We will first show that for any such $\mathcal{C}'$ and any observable $M, \|M\| \leq 1$, we have that:

$$\left( \sum_{\rho \in \mathcal{C}'} |\mathsf{Tr}(M(\rho - \sigma)| \right)^2 \leq |\mathcal{C}'|^2 \gamma(\mathcal{C}', \sigma). \tag{24}$$

---

[5]Note that if $\mathbb{E}_{\rho \sim \mathcal{C}}[\rho] \in \mathcal{C}$, then $d \to \infty$, since the average is not distinguishable from each state in $\mathcal{C}$ by any measurement. This edge case happens for example for $|\mathcal{C}| = 1$.

To that end, observe that:

$$\left(\sum_{\rho\in\mathcal{C}'}|\mathsf{Tr}(M(\rho-\sigma)|\right)^2 = \left(\sum_{\rho\in\mathcal{C}'}|\mathsf{Tr}(M\hat{\rho}\sigma)|\right)^2 = \left[\mathsf{Tr}\left(\sqrt{\sigma}M\sum_{\rho\in\mathcal{C}'}\mathsf{sign}(\mathsf{Tr}(M\hat{\rho}\sigma)\hat{\rho}\sqrt{\sigma}\right)\right]^2$$
$$\leq \mathsf{Tr}(\sigma M^2)\mathsf{Tr}\left(\left[\sum_{\rho\in\mathcal{C}'}\mathsf{sign}(\mathsf{Tr}(M\hat{\rho}\sigma)\hat{\rho}\right]^2\sigma\right),$$

$$(25)$$

where the above follows from Cauchy-Schwartz inequality. Since $\|M\|\leq 1$, we have that $\mathsf{Tr}(\sigma M^2)\leq 1$ and also that:

$$\mathsf{Tr}\left(\left[\sum_{\rho\in\mathcal{C}'}\mathsf{sign}(\mathsf{Tr}(M\hat{\rho}\sigma)\hat{\rho}\right]^2\sigma\right) = \sum_{\rho_1,\rho_2\in\mathcal{C}'}\mathsf{sign}(\mathsf{Tr}(M\hat{\rho}_1\sigma))\mathsf{sign}(\mathsf{Tr}(M\hat{\rho}_2\sigma))\mathsf{Tr}\left[\hat{\rho}_1\hat{\rho}_2\sigma\right]$$

$$(26)$$

$$\leq |\mathcal{C}'|^2\gamma(\mathcal{C}',\sigma).$$

We show the claim by upper-bounding the $\kappa_\tau$-$\mathsf{frac}(\mu,\sigma)$ for $\mu$ uniform over some subset $\mathcal{C}_0\subseteq\mathcal{C}$ by $\kappa_\tau^\gamma$-$\mathsf{frac}(\mathcal{C}_0,\sigma)$. Recall that for a distribution $\mu$ over quantum states, we have that:

$$\kappa_\tau(\mu,\sigma) = \max_{M,\|M\|\leq 1}\left\{\Pr_{\rho\sim\mu}\left[|\mathsf{Tr}(M(\rho-\sigma)|>\tau\right]\right\}$$

$$(27)$$

For a uniform distribution $\mu_{\mathcal{C}_0}$ over $\mathcal{C}_0\subseteq\mathcal{C}$, this gives:

$$\kappa_\tau(\mu_{\mathcal{C}_0},\sigma) = \max_{M,\|M\|\leq 1}\frac{1}{|\mathcal{C}_0|}\sum_{\rho\in\mathcal{C}_0}\delta\left[|\mathsf{Tr}(M(\rho-\sigma)|>\tau\right],$$

$$(28)$$

where $\delta[x]=1$ if the clause $x$ is true, and 0 otherwise. From here onwards, fix $M$ to be the operator that maximizes the above expression. Let $\mathcal{C}'\subseteq\mathcal{C}_0$ to be the largest subset of $\mathcal{C}_0$, such that $|\mathsf{Tr}(M(\rho-\sigma))|>\tau$ for all $\rho\in\mathcal{C}'$. Then $\sum_{\rho\in\mathcal{C}'}|\mathsf{Tr}(M(\rho-\sigma)|>|\mathcal{C}'|\tau$. Along with $\sum_{\rho\in\mathcal{C}'}\delta[|\mathsf{Tr}(M(\rho-\sigma)|>\tau]=|\mathcal{C}'|$ this implies that:

$$\sum_{\rho\in\mathcal{C}_0}\delta\left[|\mathsf{Tr}(M(\rho-\sigma)|>\tau\right] \leq \max_{\mathcal{C}'\subseteq\mathcal{C}_0}\left[|\mathcal{C}'|\delta\left(\sum_{\rho\in\mathcal{C}'}|\mathsf{Tr}(M\rho-\sigma)|>|\mathcal{C}'|\tau\right)\right].$$

$$(29)$$

Combining this with Eq. (28), this gives:

$$\kappa_\tau(\mu_{\mathcal{C}_0},\sigma) \leq \max_{\mathcal{C}'\subseteq\mathcal{C}_0}\left\{\frac{|\mathcal{C}'|}{|\mathcal{C}_0|}\left|\sum_{\rho\in\mathcal{C}'}|\mathsf{Tr}(M(\rho-\sigma)|>|\mathcal{C}'|\tau\right\}.$$

$$(30)$$

Using $\left(\sum_{\rho\in\mathcal{C}'}|\mathsf{Tr}(M(\rho-\sigma)|\right)^2 \leq |\mathcal{C}'|^2\gamma(\mathcal{C}',\sigma)$ implies

$$\kappa_\tau-\mathsf{frac}(\mu_{\mathcal{C}_0},\sigma) \leq \min_{\mathcal{C}'\subseteq\mathcal{C}}\kappa_{\tau^2}^\gamma-\mathsf{frac}(\mathcal{C}',\sigma).$$

$$(31)$$

Hence we have that

$$\mathsf{QSD}_\tau(\mathcal{C},\sigma) = \sup_\mu(\kappa_\tau(\mu,\sigma)^{-1}) \geq (\kappa_\tau(\mu_{\mathcal{C}_0},\sigma)^{-1}) \geq \max_{\mathcal{C}'\subseteq\mathcal{C}}(\kappa_{\tau^2}^\gamma-\mathsf{frac}(\mathcal{C}',\sigma)^{-1}) = \mathsf{QAC}_{\tau^2}(\mathcal{C},\sigma).$$

This proves the lower bounds in the theorem statement. $\square$

In many of the bounds to be proved in the following sections we consider converting a learning problem to a decision problem $\mathcal{C}$ versus $\sigma$ where $\min_{\rho\in\mathcal{C}}d_{\mathsf{Tr}}(\rho,\sigma)\geq\zeta$, where $\zeta$ is some constant. For large enough $\tau$ lemma 1 may no longer hold. Fixing an approximation error $\varepsilon$, lemma 1 then holds if $\zeta-2\varepsilon>2\tau$. Note that the left hand side is some constant and we implicitly assume this upper bound on $\tau$ in the following proofs. This is without loss of generality: the existence of a QSQ algorithm with tolerance $\tau$ greater than or equal to $\zeta-2\varepsilon$ then further implies that one exists for all tolerances of smaller value. That is, smaller tolerance cannot increase the query/time complexity. As many of the results are asymptotics, requiring that $\tau$ be at most some constant does not change the results where $\tau$ appears in the complexity.

# 4 Separations between statistical and entangled measurements

In this section we prove our main theorem separating noisy entangled QPAC learning and QSQ learning, and next show that for a "small" circuit one can witness such an exponential separation.

## 4.1 Separation between QSQ and QPAC with classification noise

In this section we prove our main theorem. Consider the class of function states

$$\mathcal{C} = \left\{ |\psi_A\rangle = \frac{1}{\sqrt{2^n}} \sum_{x \in \{0,1\}^n} |x, x^\top A x \ (\text{mod } 2)\rangle : A \in \mathbb{F}_2^{n \times n} \right\}.$$

The sample complexity of learning this class in the following models is given as follows

1. Entangled measurements: $\Theta(n)$
2. Separable measurements: $\Theta(n^2)$
3. Statistical query learning: $\Omega(\tau^2 \cdot 2^{n/2})$ making $\mathsf{Qstat}(\tau)$ queries.
4. $\eta$-random classification noise: $O(\frac{n}{(1-2\eta)^2})$. The algorithm runs in time $O(n^3/(1-2\eta)^2)$.

Points $(1), (2)$ above were proved in [15] and we do not prove it here. In the following two theorems we prove points $(3), (4)$ above.

**Theorem 5.** *The concept class*

$$\mathcal{C} = \left\{ |\psi_A\rangle = \frac{1}{\sqrt{2^n}} \sum_{x \in \{0,1\}^n} |x, x^\top A x \ (\text{mod } 2)\rangle : A \in \mathbb{F}_2^{n \times n} \right\}$$

*requires* $2^{\Omega(n)}$ *many* $\mathsf{Qstat}(1/\operatorname{poly}(n))$ *to learn below error* $0.05$ *in trace distance.*

*Proof.* We prove the hardness for algorithms using Qstat queries using the variance lower bounding technique in Theorem 4. In particular, we show an exponentially small upper bound for the variance for any observable: for every $n + 1$ qubit operator $M$ such that $\|M\| \leq 1$ we have that

$$\mathsf{Var}_A(\mathsf{Tr}[M\psi_A]) = 2^{-\Omega(n)}, \tag{32}$$

where we let $\psi_A = |\psi_A\rangle\langle\psi_A|$ for notational simplicity. To apply our results linking learning and decision problems we note that

$$d_{\mathsf{Tr}}(\psi_A, \mathbb{E}_B[\rho_B]) \geq 1 - \sqrt{\mathbb{E}_B[|\langle\psi_A|\psi_B\rangle|]^2} \geq 1 - \sqrt{\frac{(2^{n(n+1)/2} - 1) \cdot 9/16 + 1}{2^{n(n+1)/2}}} \geq 1 - \sqrt{17/32},$$

where the first inequality follows from the lower bound on trace distance by fidelity [20] and the second by fact 5 and that $\langle\psi_A|\psi_B\rangle = \Pr_x[f_A(x) = f_B(x)]$. Fix $\varepsilon = 0.05$. Then lemma 1 holds if $\tau < 0.085$, which we assume without loss of generality as previously discussed[6]. Along with Theorem 4, we obtain our lower bound on the QSQ complexity of learning $\mathcal{C}$. It remains to establish Eq. (32). To this end, we need to understand

$$\mathsf{Var}_A(\mathsf{Tr}[M\psi_A]) = \mathbb{E}_A[\mathsf{Tr}[M\psi_A]^2] - (\mathbb{E}_A[\mathsf{Tr}[M\psi_A]])^2 \tag{33}$$

To do so, we decompose $\psi_A$ as follows. For every $f : \{0,1\}^n \to \{0,1\}$ let $|\psi_f\rangle = \frac{1}{\sqrt{2^n}} \sum_x |x, f(x)\rangle$ and $|\phi_f\rangle = \sum_x (-1)^{f(x)}|x\rangle$. For convenience we let $|u\rangle = \frac{1}{\sqrt{2^n}} \sum_x |x\rangle$. Then we see that

$$(\mathbb{I} \otimes H)\psi_A(\mathbb{I} \otimes H) \tag{34}$$

$$= \frac{1}{2} \sum_{x,y,a,b} (-1)^{a \cdot f(x) + b \cdot f(y)} |x, a\rangle\langle y, b| \tag{35}$$

$$= \frac{1}{2} \left( |\phi_A\rangle\langle\phi_A| \otimes |1\rangle\langle1| - \frac{1}{2}|\phi_A\rangle\langle u| \otimes |1\rangle\langle0| - \frac{1}{2}|u\rangle\langle\phi_A| \otimes |0\rangle\langle1| + \frac{1}{2}|u\rangle\langle u| \otimes |0\rangle\langle0| \right) \tag{36}$$

---

[6] The choice of $\varepsilon = 0.05$ is arbitrarily and done for readability. A similar result holds for any $\varepsilon < 1/2(1 - \sqrt{17/32})$ by the same argument.

hence we have that

$$|\psi_A\rangle\langle\psi_A| = \frac{1}{2}\Big(\underbrace{|\phi_A\rangle\langle\phi_A| \otimes |-\rangle\langle-|}_{\rho_1^A} - \underbrace{|\phi_A\rangle\langle u| \otimes |-\rangle\langle+|}_{\rho_2^A} - \underbrace{|u\rangle\langle\phi_A| \otimes |+\rangle\langle-|}_{\rho_3^A} + \underbrace{|u\rangle\langle u| \otimes |+\rangle\langle+|}_{\rho_4^A}\Big).$$

We now note that any $n+1$ qubit observable $M$ can be decomposed as $M = \sum_{a,b} M_{a,b} \otimes |a\rangle\langle b|$ where now $a,b \in \{+,-\}$. Since $\|M\| \leq 1$ we also have that $\|M_{a,b}\| \leq 1$, however the off-diagonal blocks now no longer need be Hermitian. In an abuse of notation we now discard the last qubit of $\rho_i^A$ and denote the resulting state also as $\rho_i^A$. For ease of notation we further introduce the notation $M_1 = M_{-,-}$, $M_2 = M_{-,+}$, $M_3 = M_{+,-}$, and $M_4 = M_{+,+}$ Thus, we see that $\mathsf{Tr}[M\psi_A] = \frac{1}{2}\sum_i \mathsf{Tr}[M_i \rho_i^A]$ and further the variance can be written as

$$\mathsf{Var}_A(\mathsf{Tr}[M\psi_A]) = \frac{1}{4}(\mathbb{E}_A[\sum_{i,j} \mathsf{Tr}[M_i\rho_i^A]\mathsf{Tr}[M_j\rho_j^A]] - (\mathbb{E}_A[\sum_i \mathsf{Tr}[M_i\rho_i^A]])^2) \tag{37}$$

$$= \frac{1}{4}\sum_{i,j}\Big[\mathbb{E}_A\mathsf{Tr}(M_i\rho_i^A)\cdot\mathsf{Tr}(M_j\rho_j^A) - \mathsf{Tr}(M_i\mathbb{E}_A[\rho_i^A])\cdot\mathsf{Tr}(M_j\mathbb{E}_A[\rho_j^A])\Big]. \tag{38}$$

Below we drop the factor of $1/4$ and bound the magnitude of each term for $i,j \in [4]$. As we will show, each term must be exponentially small. To do so, we use the following facts which will be proven later.

**Fact 6.** *We have the following*

$$\mathbb{E}_A[|\phi_A\rangle] = |0^n\rangle/\sqrt{2^n}, \qquad \mathbb{E}_A[|\phi_A\rangle^{\otimes 2}] = |\Phi^+\rangle/\sqrt{2^n}, \qquad \mathbb{E}_A[|\phi_A\rangle\langle\phi_A|] = \mathbb{I}/2^n,$$

$$\mathbb{E}_A[|\phi_A\rangle \otimes \langle\phi_A|] = \frac{1}{2^n}\sum_x |x\rangle \otimes \langle x|, \quad \mathbb{E}_A[|\phi_A\rangle \otimes |\phi_A\rangle\langle\phi_A|] = \frac{1}{2^{3n/2}}|0\rangle \otimes |0\rangle\langle 0|,$$

$$\mathbb{E}_A[|\phi_A\rangle\langle\phi_A| \otimes |\phi_A\rangle] = \frac{1}{2^{3n/2}}(\sum_x |x\rangle\langle x| \otimes |0\rangle + |x\rangle\langle 0| \otimes |x\rangle + |0\rangle\langle x| \otimes |x\rangle - 2|0\rangle\langle 0| \otimes |0\rangle),$$

$$\mathbb{E}_A[|\phi_A\rangle\langle\phi_A|^{\otimes 2}] = \frac{1}{4^n}(\mathbb{I} + \mathsf{SWAP}) + \frac{1}{2^n}|\Phi^+\rangle\langle\Phi^+| - \frac{2}{4^n}\sum_x |x,x\rangle\langle x,x|,$$

*where* $\mathsf{SWAP}$ *swaps two $n$-qubit registers via* $\mathsf{SWAP}|\psi\rangle \otimes |\phi\rangle = |\phi\rangle \otimes |\psi\rangle$ *and* $|\Phi^+\rangle = 2^{-n/2}\sum_x |x,x\rangle$. *is the EPR state of $2n$ qubits.*

First note that $\rho_4^A$ does not depend on $A$. Thus, for all $i \in [4]$ we have that

$$\mathbb{E}_A[\mathsf{Tr}[M_i \otimes M_4(\rho_i^A \otimes \rho_4^A)]] = \mathsf{Tr}[M_i\mathbb{E}_A[\rho_i^A]]\mathsf{Tr}[M_4\mathbb{E}_A[\rho_4^A]]. \tag{39}$$

The contribution to the variance from these cases equals 0. We are left with analyzing $i,j \in [3]$. We analyze these cases now separately. **Case $i = j = 1$.** Using Fact 6 above, we get that

$$\mathsf{Tr}(M_1\mathbb{E}_A[\rho_1^A]) = \mathsf{Tr}(M_1 \cdot \mathbb{E}_A[|\phi_A\rangle\langle\phi_A|]) = \mathsf{Tr}(M_1)/2^n.$$

Next, observe that

$$\mathbb{E}_A\big(\mathsf{Tr}(M_1\rho_1^A)\big)^2 = \mathsf{Tr}(M_1 \otimes M_1 \cdot \mathbb{E}_A[\rho_1^A \otimes \rho_1^A])$$

$$= \mathsf{Tr}\Big(M_1 \otimes M_1 \cdot \Big(\frac{1}{4^n}(\mathbb{I} + \mathsf{SWAP}) + \frac{1}{2^n}|\Phi^+\rangle\langle\Phi^+| - \frac{2}{4^n}\sum_x |x,x\rangle\langle x,x|\Big)\Big)$$

$$= \frac{1}{4^n}\big(\mathsf{Tr}(M_1)\big)^2 + \frac{1}{4^n}\mathsf{Tr}(M_1)^2) + \frac{1}{2^n}\langle\Phi^+|M_1|\Phi^+\rangle - \frac{2}{4^n}\sum_x M_1(x,x)^2$$

$$\leq \frac{1}{4^n}\big(\mathsf{Tr}(M_1)\big)^2 + \frac{1}{2^n} + \frac{1}{2^n},$$

where the third equality used that $\mathsf{Tr}(M_1 \otimes M_1 \cdot \mathsf{SWAP}) = \mathsf{Tr}(M_1^2)$, the fourth equality used that $\mathsf{Tr}(M_1^2) \leq 2^n$ and $\|M_1^{\otimes 2}\| \leq 1$. Implicitly we have used that $M_1$ is Hermitian and $\|M_1\| \leq 1$.

Hence we have that variance term contribution is

$$\mathbb{E}_A[\mathsf{Tr}\big(M_1\rho_1^A\big)\cdot\mathsf{Tr}\big(M_1\rho_1^A\big)] - \mathsf{Tr}(M_1\mathbb{E}_A[\rho_1^A])\cdot\mathsf{Tr}(M_1\mathbb{E}_A[\rho_1^A])$$

$$\leq \frac{1}{4^n}\big(\mathsf{Tr}(M_1)\big)^2 + \frac{2}{2^n} - \Big(\mathsf{Tr}(M_1)/2^n\Big)^2 = 2/2^n.$$

**Case $i = j = 2$.** Using Fact 6 above, we get that

$$\mathsf{Tr}(M_2\mathbb{E}_A[\rho_2^A]) = \mathsf{Tr}(M_2\cdot\mathbb{E}_A[|\phi_A\rangle\langle u|]) = \langle 0|M_2|u\rangle/\sqrt{2^n}.$$

Next note that

$$\mathbb{E}_A[\mathsf{Tr}[M_2\rho_2^A]^2] = \mathsf{Tr}[M_2\otimes M_2\,\mathbb{E}_A[\rho_2^f\otimes\rho_2^A]] \tag{40}$$

$$= \mathsf{Tr}[M_2\otimes M_2\,\mathbb{E}_A[|\phi_A\rangle^{\otimes 2}]\langle u|^{\otimes 2}] \tag{41}$$

$$= \frac{1}{\sqrt{2^n}}\langle u,u|M_2\otimes M_2|\Phi^+\rangle = \frac{1}{2^n}\sum_x\langle u|M_2|x\rangle^2 = \frac{1}{4^n}\sum_x(M_2(y,x))^2 \tag{42}$$

We now bound the norm of each of these terms individual as then, by triangle inequality, the norm of the contribution from the case is exponentially small as well.

$$|(\langle 0|M_2|u\rangle/\sqrt{2^n})| \leq \frac{1}{\sqrt{2^n}}\sqrt{\||0\rangle\|^2\,\|M_2|u\rangle\|^2} \leq \frac{1}{\sqrt{2^n}}\ , \tag{43}$$

and thus $|\mathsf{Tr}(M_2\mathbb{E}_A[\rho_2^A])| \leq \frac{1}{2^n}$. For the other term we use that $|\sum_x(M_2(y,x))^2| \leq \sum_x|\sum_y M_2(y,x)|^2$. Then we can rewrite this as $\frac{1}{2^n}\|M_2|u\rangle\|_2^2 \leq \frac{1}{2^n}\|M_2\| \leq \frac{1}{2^n}$. Thus, the contribution from this case is exponentially small. **Case $i = j = 3$.** This is the same as the case $i = j = 2$, thus the norm of this case is upper bounded by $\frac{2}{2^n}$ as well. **Case $i = 2$ and $j = 3$** Using Fact 6 note that

$$\mathsf{Tr}[M_2\otimes M_3\mathbb{E}_A[\rho_2^A\otimes\rho_3^A]] = \mathsf{Tr}[M_2\otimes M_3(\mathbb{I}\otimes|u\rangle)\mathbb{E}_A[|\phi_A\rangle\otimes\langle\phi_A|](\langle u|\otimes\mathbb{I})] \tag{44}$$

$$= \frac{1}{2^n}\sum_x\mathsf{Tr}[M_2\otimes M_3(|x\rangle\langle u|\otimes|u\rangle\langle x|)] \tag{45}$$

Now we use that $M_3 = M_2^\dagger$ to rewrite this as $\frac{1}{2^n}\sum_x|\langle u|M_2|x\rangle|^2$

$$\frac{1}{4^n}\sum_x|\langle u|M_2|x\rangle|^2 = \frac{1}{2^n}\|M_2|u\rangle\|_2^2 \leq \frac{1}{2^n} \tag{46}$$

We have already shown the subtracted terms to be exponentially small in magnitude and thus the magnitude of this case must be exponentially small. **Case $i = 1$ and $j = 2, 3$.** Here we work out $j = 2$ as the result then holds similarly for $j = 3$.

$$|\mathsf{Tr}[M_1\otimes M_2\mathbb{E}_A[\rho_1^A\otimes\rho_2^A]]| = |\mathsf{Tr}[M_1\otimes M_2\mathbb{E}_A[|\phi_A\rangle\langle\phi_A|\otimes|\phi_A\rangle](\mathbb{I}\otimes\langle u|)]| \tag{47}$$

$$= |\frac{1}{2^{3n/2}}\mathsf{Tr}[M_1\otimes M_2(\sum_x|x\rangle\langle x|\otimes|0\rangle\langle u| + |x\rangle\langle 0|\otimes|x\rangle\langle u| + |0\rangle\langle x|\otimes|x\rangle\langle u| - 2|0\rangle\langle 0|\otimes|0\rangle\langle u|)]| \tag{48}$$

We deal with each term in the trace above separately. First:

$$\frac{1}{2^{3n/2}}|\mathsf{Tr}[M_1\otimes M_2(\sum_x|x\rangle\langle x|\otimes|0\rangle\langle u|)] = \frac{1}{2^{3n/2}}|\mathsf{Tr}[M_2]\cdot\langle u|M_2|0\rangle| \leq \frac{1}{2^{n/2}}\ . \tag{49}$$

The next two terms are similar and we bound the first here (which implies the same upper bound for the second via nearly identical steps).

$$\frac{1}{2^{3n/2}}|\mathsf{Tr}[M_1\otimes M_2(\sum_x|x\rangle\langle 0|\otimes|x\rangle\langle u|)] = \frac{1}{2^n}|\langle 0,u|M_1\otimes M_2|\Phi^+\rangle| \leq \frac{1}{2^n}\ . \tag{50}$$

The last remaining term is bounded as follows:

$$\frac{1}{2^{3n/2}}|\mathsf{Tr}[M_1\otimes M_2|0\rangle\langle 0|\otimes|0\rangle\langle u|]| = \frac{1}{2^{3n/2}}|\mathsf{Tr}[M_1|0\rangle\langle 0|]|\cdot|\mathsf{Tr}[M_2|0\rangle\langle u|]| \leq \frac{1}{2^{3n/2}}\ . \tag{51}$$

Thus, the contribution from the second moments is of magnitude at most $O(2^{-n/2})$. While $|\mathsf{Tr}[M_1\mathbb{E}_A[\rho_1^A]]|$ may be large (up to 1), we also have that $|\mathsf{Tr}[M_2\mathbb{E}_A[\rho_2^A]]| \leq \frac{1}{\sqrt{2^n}}$ and thus all terms in the case are exponentially small in norm as well. We have thus shown that the norms of the contributions for each case are all exponentially small. Thus, the variance must be exponentially small as well. It remains to prove Fact 6 which we prove now.

*Proof of Fact 6.* Most of the desired expectation values stem from the following observation: for the uniform distribution over upper triangular binary matrices $A$, we have that

$$\mathbb{E}_A[(-1)^{x^\top Ax + y^\top Ay + z^\top Az}] = \mathbb{E}_A[(-1)^{\langle X+Y+Z, A\rangle}] \tag{52}$$

$$= \delta_{X+Y+Z, \mathbf{0}} \tag{53}$$

$$= \delta_{x,y}\delta_{z,0} + \delta_{x,z}\delta_{y,0} + \delta_{y,z}\delta_{x,0} - 2\delta_{x,0}\delta_{y,0}\delta_{z,0}, \tag{54}$$

where $X, Y, Z$ are defined as $xx^\top, yy^\top, zz^\top$ respectively and the second equality follows from $\mathbb{E}_z[(-1)^{\langle x,z\rangle}] = \delta_{x,z}$. Now the first three inequalities in the fact are now easy to see. Setting $y = z = 0$ we have that

$$\mathbb{E}_A[|\phi_A\rangle] = \mathbb{E}_A[\frac{1}{\sqrt{2^n}}\sum_x(-1)^{x^\top Ax}|x\rangle] = \frac{1}{\sqrt{2^n}}\sum_x \mathbb{E}_A[(-1)^{x^\top Ax}]|x\rangle = |0^n\rangle/\sqrt{2^n}\ .$$

Similarly, setting $z = 0$ yields

$$\mathbb{E}_A[|\phi_A\rangle^{\otimes 2}] = \mathbb{E}_A[\frac{1}{2^n}\sum_{x,y}(-1)^{x^\top Ax + y^\top Ay}|x,y\rangle] = \frac{1}{2^n}\sum_{x,y}\mathbb{E}_A[(-1)^{x^\top Ax + y^\top Ay}]|x,y\rangle = \frac{1}{2^n}\sum_x |x,x\rangle\ .$$

Similar reasoning implies $\mathbb{E}_A[|\phi_A\rangle\langle\phi_A|] = \mathbb{I}/2^n$, $\mathbb{E}_A[|\phi_A\rangle \otimes \langle\phi_A|] = \frac{1}{2^n}\sum_x |x\rangle \otimes \langle x|$, and

$$\mathbb{E}_A[|\phi_A\rangle\langle\phi_A| \otimes |\phi_A\rangle] = \frac{1}{2^{3n/2}}(\sum_x |x\rangle\langle x| \otimes |0\rangle + |x\rangle\langle 0| \otimes |x\rangle + |0\rangle\langle x| \otimes |x\rangle - 2|0\rangle\langle 0| \otimes |0\rangle)\ .$$

The final decomposition of $\mathbb{E}_A[|\phi_A\rangle\langle\phi_A|^{\otimes 2}]$ follows from [15, Proposition 2]. $\square$

The proof of the fact concludes the proof of the theorem. $\square$

Here we note that an alternative proof technique, similar to that used later to prove lower bounds on purity testing and learning distributions, extends this result to hold for $\varepsilon < \sqrt{7/16}$. However, we omit it here in the interest of cohesiveness.

Next, we show that $\mathcal{C}$ is efficiently learnable in QPAC even in the presence of classification noise.

**Theorem 6.** *The concept class*

$$\mathcal{C} = \left\{ |\psi_A\rangle = \frac{1}{\sqrt{2^n}}\sum_{x\in\{0,1\}^n} |x, x^\top Ax \ (mod\ 2)\rangle : A \in \mathbb{F}_2^{n\times n} \right\}$$

*can be learned in the $\eta$-random classification model, using $O(n/(1-2\eta)^2)$ copies of the noisy state and time $O(n^3/(1-2\eta)^2)$.*

*Proof.* Below, let $|\phi_f\rangle = \frac{1}{\sqrt{2^n}}\sum_x(-1)^{f(x)}|x\rangle$. In the random classification noise model, we are given copies of

$$|\psi_f\rangle = \frac{1}{\sqrt{2^n}}\sum_x |x\rangle \otimes \left(\sqrt{1-\eta}|f(x)\rangle + \sqrt{\eta}|\overline{f(x)}\rangle\right).$$

We first show that using two copies of $|\psi_f\rangle$, with probability $\geq \cdots$, we can obtain $|\psi_f\rangle^{\otimes 2}$. In order to do so, observe the following

$$(\mathbb{I} \otimes H)|\psi_n\rangle = \frac{1}{\sqrt{2^{n+1}}}\sum_{x,b} |x\rangle\left((-1)^{b\cdot f(x)}|b\rangle + \sqrt{\eta}(-1)^{b\cdot\overline{f(x)}}|b\rangle\right)$$

$$= \frac{1}{\sqrt{2^{n+1}}}\sum_{x,b}\left(\sqrt{1-\eta}(-1)^{b\cdot f(x)} + \sqrt{\eta}(-1)^{b\cdot\overline{f(x)}}\right)|x,b\rangle.$$

Now, measuring the last qubit, the probability of seeing $b = 1$ is given by

$$\left\|\frac{1}{\sqrt{2^{n+1}}}\sum_x\left(\sqrt{1-\eta}(-1)^{f(x)} - \sqrt{\eta}(-1)^{f(x)}\right)|x\rangle\right\|_2^2$$

$$= \frac{1}{2^{n+1}}\sum_x\left(\sqrt{1-\eta}(-1)^{f(x)} - \sqrt{\eta}(-1)^{f(x)}\right)^2 = \frac{1}{2}(\sqrt{1-\eta} - \sqrt{\eta})^2 =: p.$$

The post-measurement state is given by

$$\sqrt{1/p} \cdot \left(\sqrt{1-\eta} - \sqrt{\eta}\right) \cdot \frac{1}{\sqrt{2^{n+1}}} \sum_x (-1)^{f(x)}|x\rangle = |\psi_f\rangle.$$

Hence with probability exactly $p = \frac{1}{2}(1 - 2\sqrt{\eta(1-\eta)}) \leq (1-2\eta)^2$ (which holds for every $\eta \leq 1/2$), given two copies of $|\psi_f\rangle$, we can produce two copies of $|\phi_f\rangle$.

Now, if we focus on the concept class $f_A(x) = x^\top A x$. The learning algorithm first takes $O(1/(1-2\eta)^2)$ copies of $|\psi_A\rangle$ to produce two copies of $|\phi_A\rangle$. Note that the algorithm knows when it succeeded, i.e., when the measurement of the last qubit is 1, the algorithm knows that the above procedure performed the transformation $|\psi_n\rangle^{\otimes 2} \to |\psi_A\rangle^{\otimes 2}$. Now using Fact 2 we can learn $f_A$ given $O(n)$ copies of $|\psi_A\rangle$ and $O(n^3)$ time. Overall, the sample complexity and time complexity of the procedure is $O(n/(1-2\eta)^2)$ and $O(n^3/(1-2\eta)^2)$ respectively. □

### 4.2  Smallest class separation

In the previous section we saw that the concept class of quadratic functions separated QPAC from QSQ. Observe that states in this concept class can be prepared by circuits of size $O(n^2)$ and depth $O(n)$ consisting of $\{\mathsf{Had}, \mathsf{X}, \mathsf{CX}\}$ gates. A natural question is, can states prepared by *smaller* circuits also witness such a separation between QPAC and QSQ? In the theorem below we answer this in the positive, by using a simple padding argument inspired by a prior work of Hinshe et al. [12].

**Theorem 7.** *Let $\alpha \in (0,1)$ there exists a family of $n$ qubit Clifford circuits of depth $d = (\log n)^{1/\alpha}$ and size $d^2$ that requires $2^{\Omega(d)}$ Qstat queries to learn the state to error $\leq 0.05$ in trace distance.*

*Proof.* The idea is to "pad" a family of circuits with auxilliary qubits. In the previous section, from Theorem 5 we saw that the set of example states $\{|\psi_A\rangle = \frac{1}{\sqrt{2^n}} \sum_x |x, x^\top A x\rangle\}_A$, is hard to learn to trace distance 0.05. Instead of the example state $|\psi_A\rangle$ now instead consider the "padded state" $|\psi_A\rangle \otimes |0\rangle^{k(n)}$. Clearly the trace distance between $|\psi_A\rangle \otimes |0\rangle^{k(n)}$ and $|\psi_B\rangle \otimes |0\rangle^{k(n)}$ remains unchanged. Say a QSQ algorithm learns these padded states with the set of Qstat queries given by $\{M_i\}_i$. Let's decompose each $M_i$ as $M_i = \sum_{x,y \in \{0,1\}^{k(n)}} M_i^{x,y} \otimes |x\rangle\langle y|$, where $\|M_i^{x,y}\| \leq 1$ and $M_i^{x,x}$ is Hermitian. Since the auxiliary qubits are fixed, it is clear that

$$\mathsf{Tr}[M \cdot |\phi_A\rangle\langle\phi_A| \otimes |0\rangle\langle 0|^{\otimes k(n)}] = \mathsf{Tr}[M_i^{0,0}\phi_A].$$

Furthermore, we can assume without loss of generality that the algorithm always outputs a state of the form $|\varphi\rangle \otimes |0\rangle^{\otimes k(n)}$ (as otherwise we could improve $\mathcal{A}$ by requiring it to do so) . Thus, a QSQ algorithm for the padded states implies a QSQ algorithm with queries $\{M_i^{0,0}\}_i$. Say that this algorithm uses at most $t$ Qstat queries. Then Theorem 5 implies that $t \geq 2^{\Omega(n)}$. The state is now composed of $m = k(n) + n$ qubits. Pick $k = 2^{n^\alpha}$ for some $\alpha < 1$, so $m = \Theta(k(n))$ and $n = (\log k)^{1/\alpha}$. Then we have that $t \geq 2^{\Omega(\log m)^{1/\alpha}}$. To see the theorem, note that $|\psi_A\rangle \otimes |0\rangle^{k(n)}$ circuit of size $O(k(n)^2) = O(m^2)$ and depth $O(k(n)) = O(m)$. Thus, the padded states can be prepared with circuits of size $(\log m)^{2/\alpha}$ and depth $(\log m)^{1/\alpha}$. □

## 5  New upper and lower bounds on QSQ learning states

In this section we first give a couple of classes of states which can be learned in the QSQ framework before discussing lower bounds for other class of states.

### 5.1  New upper bounds

We first prove that the class of functions that are $k$-Fourier-sparse Boolean functions on $n$ bits, i.e.,

$$\mathcal{C}_1 = \{f : \{0,1\}^n \to \{0,1\} : |\mathsf{supp}(\widehat{f})| = k\}$$

can be learned in time $\mathrm{poly}(n,k)$ in the QSQ model. This generalizes the results in [21, 10], which showed that showed that parities and $O(\log n)$-juntas (which are a subset of Fourier-sparse functions) are

poly$(n)$-time learnable.[7] We observe that the quantum coupon collector problem, i.e., learnability of

$$\mathcal{C}_2 = \{S \subseteq [n] : |S| = k\}$$

considered in [23] can be implemented in QSQ. Finally, we also observe that one can learn codeword states defined in [8]: consider an $[n, k, d]_2$ linear code $\{Mx : x \in \{0,1\}^k\}$ where $G \in \mathbb{F}_2^{n \times k}$ is a rank-$k$ generator matrix of the code, $k = \Omega(n)$, and distinct codewords have Hamming distance at least $d$, then define the concept class

$$\mathcal{C}_3 = \{f_x(i) = (Gx)_i : x \in \{0,1\}^k\},$$

where $G$ is known the learning algorithm. Below we show we can learn $\mathcal{C}_3$ in the QSQ model. Prior learning protocols [8, 23, 21, 10] showed that these concept classes are learnable with quantum examples (a stronger model than QSQ) whereas here we show they are learnable in the weaker QSQ framework. Before we prove this theorem, we will use the following lemmas.

**Lemma 5.** *[24, Theorem 12] Let $k \geq 2$. The Fourier coefficients of a $k$-Fourier-sparse Boolean function $f : \{0,1\}^n \to \{-1,1\}$ are integer multiples of $2^{1-\lfloor \log k \rfloor}$.*

**Lemma 6.** *[10, Theorem 4.4] Let $f : \{-1,1\}^n \to \{-1,1\}$, $\tau \in (0,1]$. There exists a* poly$(n, 1/\tau, \ell)$-*time quantum statistical learning algorithm that with high probability outputs* $U = \{T_1, \ldots, T_\ell\} \subseteq [n]$ *such that: (i) if $|\widehat{f}(T)| \geq \tau$, then $T \in U$; and (ii) if $T \in U$, then* $|\widehat{f}(T)| \geq \tau/2$.

**Theorem 8.** *The concept classes $\mathcal{C}_1, \mathcal{C}_2, \mathcal{C}_3$ defined above can be learned in the* QSQ *model.*

*Proof.* We first give a learning algorithm for $\mathcal{C}_1$. For every $f \in \mathcal{C}_1$, observe that it's Fourier coefficients satisfy $|\widehat{f}(S)| \geq 1/k$ by Lemma 5. We can now use Lemma 6 to collect *all* the non-zero Fourier coefficients in time poly$(n, 1/\tau, k)$ in the QSQ model. Call these non-zero coefficients $S_1, \ldots, S_k$. Next, we learn all these Fourier coefficients up to error $\varepsilon/k$ using Stat queries: for $i \in [k]$, let $\phi(x, b) = b \cdot (-1)^{S_i \cdot x}$ for all $x \in \{0,1\}^n, b \in \{0,1\}$, hence $\mathbb{E}_x[\phi(x, f(x))] = \mathbb{E}_x[f(x) \cdot (-1)^{S_i \cdot x}] = \widehat{f}(S_i)$. Overall this takes time $O(k)$. Once we obtain all these approximations $\{\alpha_i\}_{i \in [k]}$, we output the function $g(x) = \mathsf{sign}\left(\sum_{i \in [k]} \alpha_i \cdot \chi_{S_i}(x)\right)$ for every $x \in \{0,1\}^n$. Using the same reasoning as in [10, Eq. (7)] it is not hard to see that $g$ is $\varepsilon$-close to $f$ (i.e., $\Pr_x[g(x) = f(x)] \geq 1 - \varepsilon$).

We next give a learning algorithm for $\mathcal{C}_2$. Let $S \subseteq [n]$ of size $k$. Given copies of $\frac{1}{\sqrt{k}} \sum_{i \in S} |i\rangle$, learn $S$. We now show how to learn $S$ in QSQ using $k \log n$ Qstat queries. Let $M_1 = \sum_{i=1}^{n/2} |i\rangle\langle i|$. This satisfies $\|M_1\| \leq 1$ and $M_1$ can be implemented using a poly$(n)$-sized circuit. Observe that

$$\langle \psi | M_1 | \psi \rangle = \frac{1}{k} \sum_{q,q' \in S} \sum_{i \in [n/2]} [q = i = q'] = \frac{|[n/2] \cap S|}{k}$$

which is at least $1/k$ if and only if there is an $i \in [n/2] \cap S$. So if we do a Qstat query with $M_1$ and tolerance $1/(2k)$, the learning algorithm learns if there is an $i \in [n/2]$ such that $i \in S$. Repeat this using a binary search and we will eventually find one element in $S$ using $O(\log n)$ Qstat queries. Repeat this to find all the elements in $S$, so the overall complexity is $O(k \log n)$.

We next give a learning algorithm for $\mathcal{C}_3$. Consider the QSQ queries $M_j = |e_j\rangle\langle e_j| \otimes |0\rangle\langle 0|$ and $\tau = 1/(2n)$. Then observe that

$$\langle \psi_x | M_j | \psi_x \rangle = [e_j^\top Mx = 0]/n,$$

which equals $1/n$ if $e_j^\top Mx = 0$ and $0$ otherwise, so with tolerance $1/(2n)$, we can learn which is the case. Since $G$ is the generator matrix of a good code, i.e., $G$ has rank $k$, there are $k$ linearly independent rows in $G \in \mathbb{F}_2^{n \times k}$ (say they are $G^{i_1}, \ldots, G^{i_k}$). The learning algorithm can perform these Qstat measurements for all $M_{i_1}, \ldots, M_{i_k}$ in order to learn $G^{i_1}x, \ldots, G^{i_k}x$. Since $G^i$s are linearly independent, these $k$ linearly independent constraints on $x$ suffice to learn $x$. $\quad\square$

---

[7]We remark that the same proof also shows that $k$-term DNF formulas are learnable: for every $g \in \mathcal{C}_1$, there exists $S$ s.t. $|\widehat{g}(S)| \geq 1/k$ and the proof of Theorem 8 can identify such an $S$ using QSQ queries and then one can use the algorithm of Feldman [22] for learning the unknown DNF formulas.

We next observe that the set of trivial states, i.e., states $|\psi\rangle = C|0^n\rangle$ where $C$ is a constant-depth $n$-qubit circuits, can be learned in polynomial time in the QSQ model. An open question of this work, and also the works of [12, 13], is if we can learn the distribution $P_C = \{\langle x|\psi\rangle^2\}_x$ using *classical* SQ queries. The theorem below shows that if we had direct access to $|\psi\rangle$, one can learn the state and the corresponding distribution $P_C$, using QSQ queries. In the next section we show that once the depth $d = \omega(\log n)$, these states are hard for QSQ queries as well.

**Theorem 9.** *The class of $n$-qubit trivial states can be learned up to trace distance $\leq \varepsilon$ using* $\mathrm{poly}(n, 1/\varepsilon)$ Qstat *queries with tolerance* $\mathrm{poly}(\varepsilon/n)$.

*Proof.* Say that the circuit depth is $d$. Via theorem 4 of [25] it is sufficient to tomography all $D := 2^d$-body reduced density matrices up to precision $\frac{\varepsilon^2}{4n}$ with respect to trace distance. Thus, it is sufficient to show that such a tomography can be accomplished with Qstat queries. This can be achieved by simply querying all $4^D - 1$ non-identity Pauli strings acting on a party of size $D$ and reconstructing the state as $\hat{\rho} = \frac{1}{2^D}(\mathbb{I} + \sum_x \alpha_x P_x)$, where $P_x$ is a Pauli string and $\alpha_x$ is the response upon querying $P_x$. The Hilbert-Schmidt distance between the resulting state and the true state $\rho$ must satisfy $\|\rho - \hat{\rho}\|_{HS} \leq 2^D \tau$, where $\tau$ is the Qstat tolerance. In general, $d_{\mathsf{Tr}}(\rho, \hat{\rho}) \leq 2^{D/2-1}\sqrt{\|\rho - \hat{\rho}\|_{HS}}$ [26]. Thus, $d_{\mathsf{Tr}}(\rho, \hat{\rho}) \leq 2^{D-1}\sqrt{\tau}$. Taking $\tau \leq \frac{1}{2^{2D-2}}\frac{\varepsilon^4}{16n^2} = O(\frac{\varepsilon^4}{n^2})$ yields a tomography with the desired precision. There are $\binom{n}{D} = O(n^D)$ such reduced density matrices. For each one, we require a constant number of Qstat queries, each requiring $O(D)$ gates. Thus, the overall complexity is $O(n^D) \in \mathrm{poly}(n)$ for both query and time complexity. $\qquad\square$

## 5.2 Hardness of testing purity

**Theorem 10.** *Let $\mathcal{A}$ be an algorithm that upon the input of a quantum state $\rho$ outputs an estimate, to accuracy $1/3$, of the purity of $\rho$ with high probability using* $\mathsf{Qstat}(\tau)$. *Then $\mathcal{A}$ must make at least* $2^{\Omega(\tau^2 \cdot 2^n)}$ *such queries.*

*Proof.* Using $\mathcal{A}$ one could solve the many-vs-one problem of $\mathcal{C} = \{U|0\rangle\langle 0|U^\dagger \mid U \in \mathcal{U}(2^n)\}$ (where $U$ is drawn from the Haar measure) versus $\sigma = \frac{1}{2^n}\mathbb{I}$. We prove that this decision problem is hard via a concentration of measure argument similar to the variance method. Note that $\mathbb{E}[U|0\rangle\langle 0|U^\dagger] = \frac{1}{2^n}\mathbb{I}$. Upon querying an observable $M$, consider the adversarial response of $\frac{1}{2^n}\mathsf{Tr}[M]$. By Levy's lemma 2, most Haar random states cannot deviate much from this average. For our purposes, we are concerned with functions of the form $f(|\psi\rangle) = \mathsf{Tr}[M|\psi\rangle\langle\psi|]$ where $\|M\| \leq 1$. We immediately observe that such $f$'s have Lipschitz constant 2 [27]. By Levy's lemma 2 we thus have that

$$\Pr[|\mathsf{Tr}[MU|0\rangle\langle 0|U^\dagger] - \frac{1}{2^n}\mathsf{Tr}[M]| > \tau] \leq 2\exp(-\frac{2^{n+1}\tau^2}{36\pi^3}) \tag{55}$$

To conclude we note that Levy's lemma directly lower bounds $\mathsf{QQC}_\tau$ (in a manner similar to that of the variance lower bound). Recall that $\mathsf{QQC}_\tau(\mathcal{C}, \sigma)$ is the smallest integer $d$ such that there exists a distribution $\eta$ over Qstat queries $M$ such that $\forall \rho \in \mathcal{C} : \Pr_{M\sim\eta}[|\mathsf{Tr}[M(\rho - \sigma)]| > \tau] \geq 1/d$. From this definition we have that

$$\frac{1}{d} \leq \Pr_{M\sim\eta}\Pr_{\rho\sim\mu}[|\mathsf{Tr}[M(\rho - \sigma)]| > \tau] \leq 2\exp(-\frac{2^{n+1}\tau^2}{36\pi^3}). \tag{56}$$

Thus, $\mathsf{QQC}_\tau(\mathcal{C}, \sigma) \geq 2^{\Omega(\tau^2 2^n)}$. Finally, observe that to succeed on at least a set of measure 0.99, any deterministic algorithm thus requires $2^{\Omega(\tau^2 2^n)}$ Qstat queries. Thus, via Yao's principle the randomized complexity is $2^{\Omega(\tau^2 2^n)}$ for any algorithm that succeeds with probability at least 0.99. $\quad\square$

## 5.3 Hardness of learning Coset states

One of the great successes of quantum computing is solving the hidden subgroup problem for Abelian groups, one such example is that of Shor's factoring algorithm. In this problem, we are given query access to a function $f$ on a group $G$ such that there is some subgroup $H \leq G$ satisfying $f$ is constant every left coset of $H$ and is distinct for different left cosets of $H$. How many queries to $f$ suffice to learn $H$? When $G$ is a finite abelian group, $H$ can be efficiently determined via separable quantum algorithms. One approach which is often used to analyze the general Hidden subgroup problem is the *standard* approach, which we describe now [28]:

1. Prepare the superposition $\frac{1}{\sqrt{|G|}} \sum_{g \in G} |g\rangle \otimes |0\rangle$ via a Fourier transform over the group $G$.

2. Use a single query to prepare the superposition state $\frac{1}{\sqrt{|G|}} \sum_{g \in G} |g\rangle \otimes |f(g)\rangle$.

3. Measure the second register and obtain a superposition over elements in some coset with representative $g'$. That is, the algorithm can be viewed as having the state $\rho_H = \sum_{g'} |\psi_{g'H}\rangle\langle\psi_{g'H}|$ where $|\psi_{g'H}\rangle = \frac{1}{\sqrt{|H|}} \sum_{g \in g'H} |g\rangle$.

4. Again apply a quantum Fourier transform and measure the state to obtain an element $g \in H^\perp$, where $H^\perp = \{g \in G | \chi_g(H) = 1\}$.

Repeating the above procedure $\tilde{O}(\log |G|)$ times yields a generating set for $H^\perp$ with high probability, allowing one to reconstruct $H$ as well. In fact observe that the above algorithm works even if one just makes *separable* measurements. The state $\rho_H$ in step (3) of the algorithm above is called a *coset state*. Here, we show that solving the Hidden subgroup problem for even Abelian groups is hard when the learning algorithm has access only to QSQ queries.

Consider the additive group $G = \mathbb{Z}_2^n$. In Simon's problem, a version of the hidden subgroup problem on $\mathbb{Z}_2^n$, the hidden subgroups are of the form $H = \{0, s\}$. While solving Simon's problem is easy via separable quantum measurements, it cannot be readily replicated via Qstat queries. Intuitively, every $y$ in the orthogonal complement of $s$ is equally likely to be observed upon a computational basis measurement. To see this, note that after discarding the register containing the function value, the resulting mixed states are $\rho_s = \frac{1}{2^{n-1}} \sum_{\overline{x}} |\overline{x}\rangle\langle\overline{x}|$, where $\overline{x}$ is a coset representative and $|\overline{x}\rangle\langle\overline{x}|$ is the projector onto the corresponding coset. Thus, accurately simulating this measurement with Qstat queries requires exponentially small tolerance $\tau$. The following theorem formalizes this notion.

**Theorem 11.** *Solving the hidden subgroup problem for the Abelian group $\mathbb{Z}_2^n$ with Qstat queries of the form $M = M' \otimes I$ requires $\Omega(\tau^2 \cdot 2^n)$ many such queries to succeed with high probability.*

*Proof.* We prove the theorem via a bound on $\mathsf{QAC}_\tau(\mathcal{C}, \sigma)$ where $\sigma = \frac{1}{2^n} \mathbb{I}$. Say that $\mathcal{A}$ is an algorithm which solves the hidden subgroup problem with high probability using Qstat queries of the form $M = M' \otimes I$. Thus, the queries $\{M_i\}_i$ used by $\mathcal{A}$ imply the existence of queries $\{M_i'\}_i$ where $M_i' \in \mathbb{C}^{2^n} \times \mathbb{C}^{2^n}$ which suffice to identify the coset states $\rho_H = \frac{1}{|H|} \sum_{\overline{x}} |\overline{x}\rangle\langle\overline{x}|$, where $\overline{x}$ denotes a coset and $|\overline{x}\rangle\langle\overline{x}|$ the projector onto this coset.

Consider the subset $\mathcal{C}_0 \subset \mathcal{C}$ of coset states of subgroups of the form $H_s = \{0, s\}$. For such a subgroup $H_s$ the corresponding coset state is $\rho_s = \frac{1}{2^{n-1}} \sum_{\overline{x}} |\overline{x}\rangle\langle\overline{x}|$, where $\{\overline{x}\}$ are a set of $2^{n-1}$ coset representatives and $|\overline{x}\rangle = \frac{1}{\sqrt{2}}(|x\rangle + |x \oplus s\rangle)$. If $f$ is a constant function, then $\rho_H = \frac{1}{2^n} I$. Thus, the correctness of $\mathcal{A}$ implies the existence of a QSQ algorithm that can solve the decision problem of $\{\rho_{H=\{0,s\}}\}_H$ versus $\sigma = \frac{1}{2^n} \mathbb{I}$.

For such a decision problem, $\hat{\rho}_s = 2^n \rho_s - \mathbb{I}$ and $\mathsf{Tr}[\hat{\rho}_s \hat{\rho}_{s'} \sigma] = 2^n \mathsf{Tr}[\hat{\rho}_s \hat{\rho}_{s'}] - 1$. Let $s = s'$. Then $\mathsf{Tr}[\rho_s^2] = 2^{-2(n-1)}$ and $\mathsf{Tr}[\hat{\rho}_s^2 \sigma] = 1$. Now instead consider when $s \neq s'$. For every coset $\overline{x}$ of $H_s$ there exist two cosets $\overline{y}_1$ and $\overline{y}_2$ of $H_{s'}$ with a non-empty intersection (of exactly one element) with $\overline{x}$. Thus, we have that $\mathsf{Tr}[|\overline{x}\rangle\langle\overline{x}||\overline{y}_1\rangle\langle\overline{y}_1|] = \mathsf{Tr}[|\overline{x}\rangle\langle\overline{x}||\overline{y}_2\rangle\langle\overline{y}_2|] = \frac{1}{4}$ and

$$\mathsf{Tr}[\hat{\rho}_s \hat{\rho}_{s'} \sigma] = 2^n \mathsf{Tr}[\rho_s \rho_{s'}] - 1 = \frac{2^n}{2^{2(n-1)}} \sum_{\overline{x}, \overline{y}} |\langle\overline{x}|\overline{y}\rangle| - 1 = \frac{2^n}{2^{2(n-1)}} \sum_{\overline{x}} \frac{1}{2} - 1 = 0 . \tag{57}$$

For any subset $\mathcal{C}' \subseteq \mathcal{C}_0$ we thus have that $\gamma(\mathcal{C}', \sigma) = \frac{1}{|\mathcal{C}'|}$. If $|\mathcal{C}'| < \frac{1}{\tau}$ then $\gamma(\mathcal{C}', \sigma) > \tau$. Note that $|\mathcal{C}_0| = 2^n - 1$ and thus $\kappa_\tau^\gamma - \mathsf{frac}(\mathcal{C}_0, \sigma) = \Theta(\frac{1}{\tau 2^n})$ and $\mathsf{QAC}_\tau(\mathcal{C}, \sigma) = \Omega(\tau \cdot 2^n)$. Via Theorem 4 we thus have that $\mathsf{QQC}_\tau(\mathcal{C}, \sigma) \geq \mathsf{QAC}_{\tau^2}(\mathcal{C}, \sigma) = \Omega(\tau^2 \cdot 2^n)$. $\square$

Thus, any QSQ algorithm for solving the hidden subgroup problem on $\mathbb{Z}_2^n$ must depend non-trivially on the register holding the function value. This is in contrast to the standard Fourier sampling method which has no dependence on the function register.

**Remark 1.** *The average correlation argument above also implies that learning coset state below trace distance $\frac{1}{2}$ with high probability requires $\Omega(\tau^2 \cdot 2^n)$ Qstat queries of tolerance $\tau$.*

*Proof.* Note that the trace distance between $\rho_H$ for $H = \{0, s\}$ and $\frac{1}{2^n}\mathbb{I}$ is $\frac{1}{2}$. Thus, via lemma 2, $QSQ_\tau^{1/2,\delta}(\mathcal{C}) \geq \mathsf{QQC}_\tau(\mathcal{D}, \frac{1}{2^n}\mathbb{I})$, where $\mathcal{D} = \{\rho_H | H = \{0, s\}\}$. From section 3 we know that $\mathsf{QQC}_\tau(\mathcal{D}, \frac{1}{2^n}\mathbb{I}) \geq (1 - 2\delta)\mathsf{QAC}_{\tau^2}(\mathcal{C}, \frac{1}{2^n}\mathbb{I})$. The average correlation argument above yields that $\mathsf{QAC}_{\tau^2}(\mathcal{C}, \frac{1}{2^n}\mathbb{I}) = \Omega(\tau^2 \cdot 2^n)$, thus proving the claim. $\square$

### 5.4 Hardness of shadow tomography

In [29] the authors derive lower bounds on the sample complexity of shadow tomography using separable measurements. Recall that in shadow tomography, given copies of $\rho$, the goal of a learner is to predict the expectation value $\mathsf{Tr}[O_i\rho]$ of a collection of known observables $\{O_i\}_i$ up to error $\varepsilon$. To prove these lower bounds the authors construct a many-vs-one decision task where $\sigma = \mathbb{I}/2^n$ and

$$\mathcal{C} = \left\{\rho_i = \frac{\mathbb{I} + 3\varepsilon O_i}{2^n}\right\}. \tag{58}$$

Assuming that $\mathsf{Tr}[O_i] = 0$ and $\mathsf{Tr}[O_i^2] = 2^n$ for all $O_i$, then an algorithm which solves the shadow tomography problem with high probability also solves the decision problem. Thus, a lower bound on the latter is also a lower bound on the sample complexity of shadow tomography.

**Theorem 12.** *Any algorithm that uses* $\mathsf{Qstat}(\tau)$ *queries and predicts* $\mathsf{Tr}[P\rho]$ *up to error* $\varepsilon$ *for all non-identity Pauli strings* $P$ *with high probability requires* $\Omega(\tau^2 \cdot 2^{2n}/\varepsilon^2)$ *queries.*

*Proof.* We prove the theorem via a bound on $\mathsf{QAC}_\tau(\mathcal{C}, \sigma)$ where $\sigma = \frac{1}{2^n}\mathbb{I}$. For convenience we label the states from the many-vs-one decision task as $\rho_i$ where $i \in [4^n - 1]$. For such a $\sigma$ we further have that $\mathsf{Tr}[\hat{\rho}_i\hat{\rho}_j\sigma] = 2^n\mathsf{Tr}[\rho\rho'] - 1$. Via the orthogonality of Pauli strings, $\mathsf{Tr}[\hat{\rho}_i\hat{\rho}_j\sigma] = 9\varepsilon^2\delta_{i,j}$. For any subset $\mathcal{C}' \subseteq \mathcal{C}$ we thus have that $\gamma(\mathcal{C}', \sigma) = \frac{9\varepsilon^2}{|\mathcal{C}'|}$. If $|\mathcal{C}'| < \frac{9\varepsilon^2}{\tau}$ then $\gamma(\mathcal{C}', \sigma) > \tau$. Thus, $\kappa_\tau^\gamma - \mathsf{frac}(\mathcal{C}, \sigma) = \Theta(\varepsilon^2 \cdot \tau^{-1} \cdot 2^{2n})$ and $\mathsf{QAC}_\tau(\mathcal{C}, \sigma) = \Omega(\frac{\tau \cdot 2^{2n}}{\varepsilon^2})$. Via Theorem 4 we know that $\mathsf{QQC}_\tau(\mathcal{C}, \sigma) \geq \mathsf{QAC}_{\tau^2}(\mathcal{C}, \sigma) = \Omega(\tau^2 \cdot 2^{2n}/\varepsilon^2)$. $\square$

We remark that when using separable measurements, the result of Chen et al. [29] showed that $O(2^n)$ many copies of $\rho$ suffice for the task above, whereas our lower bound shows that $\Omega(4^n)$ copies are necessary if one only has access to $\mathsf{Qstat}$ measurements (in other words, obtaining the expectation value with *every* Pauli observable is necessary). This implies that measurements not just their statistics play a non-trivial role in shadow tomography.

### 5.5 Learning quantum biclique states

An influential work of Feldman et al. [19] considers the planted biclique problem. The goal here is to learn the class of distributions each indexed by subsets $S \subseteq \{1, 2 \dots, n\}$. For every $S$, the distribution $D_S$ is defined as follows

$$D_S(x) = \begin{cases} \frac{k/n}{2^{n-k}} + \frac{1-k/n}{2^n} & x \in 1_S \times \{0, 1\}^{n-k} \\ \frac{1-k/n}{2^n} & x \notin 1_S \times \{0, 1\}^{n-k}, \end{cases}$$

where above $1_S \times \{0, 1\}^{n-k}$ is the set $\{x \in \{0, 1\}^n : x_S = 1_S\}$.

A natural way of generalizing problems over distributions to quantum statistical queries is to consider coherent encodings of distributions, i.e., for a given distribution $D$ over $X$, we define a quantum state $|\psi\rangle = \sum_x \sqrt{D(x)}|x\rangle$. Classical $\mathsf{Stat}$ queries then correspond to $\mathsf{Qstat}$ with *diagonal* observables and a natural question is, how much can coherent examples help?

In what follows, we first show that for the task of distinguishing two coherent encodings, there can be at most a quadratic gap between the precision that is tolerated by $\mathsf{Qstat}$ and $\mathsf{Stat}$ queries. We use this to show that, for some choice of parameters, there are large gaps between the classical and quantum statistical query complexity of the $k$-biclique problem. We demonstrate below that $\mathsf{Qstat}$ measurements can help significantly in certain regimes of tolerance.

**Lemma 7.** *For large enough* $n$ *and* $k \geq 2\log n$, *the* $k$-planted biclique problem with coherent encodings can be solved with statistical quantum algorithm that makes at most $\binom{n}{k}$ $\mathsf{Qstat}\left(\sqrt{k/n}\right)$ queries, but cannot be solved by any algorithm that makes $\mathsf{Stat}\left(\sqrt{k/n}\right)$ queries.

*Proof.* First observe that $d_{\mathsf{Tr}}(|\psi_S\rangle, |+^n\rangle) = \sqrt{1 - |\langle\psi_S|+^n\rangle|^2}$, and

$$\langle +^n|\psi_S\rangle = \left(\sqrt{\frac{k}{n} + \frac{1-k/n}{2^k}} - \sqrt{\frac{1-k/n}{2^k}}\right)\frac{1}{\sqrt{2^k}} + \sqrt{1 - \frac{k}{n}}. \tag{59}$$

Define $|\phi\rangle = (|+\rangle + |\psi_S\rangle)/\sqrt{2 + 2\langle\psi_S|+\rangle}$ and $|\phi^{\mathsf{T}}\rangle (|+\rangle - |\psi_S\rangle)/\sqrt{2 - 2\langle\psi_S|+\rangle}$. The optimal distinguishing Qstat query between $|+^n\rangle$ and $|\psi_S\rangle$ is the difference between projectors on the state $|P_S\rangle = \frac{|\phi\rangle + |\phi^{\mathsf{T}}\rangle}{\sqrt{2}}$ and its orthogonal complement in the span of $|+^n\rangle$ and $|\psi_S\rangle$ (see for example [30, Theorem 3.4]). Call this measurement $M_S$ and notice that it is implementable by a $k$-qubit controlled rotation. A (possibly inefficient) quantum algorithm for detecting the planted clique would query Qstat$(\tau)$ oracle with $M_S$ for every subset $S \subseteq [n]$ of cardinality $|S| = k$. From Lemma 4 and optimality of the measurement, we know that $|\mathsf{Tr}(M_S(\psi_S - \psi_0))| = 2d_{\mathsf{Tr}}(\psi_S, \psi_0)$. It follows that as long as $\tau \le d_{\mathsf{Tr}}(|\Psi_S\rangle, |+^n\rangle)$, such algorithm succeeds.

We now bound $d_{\mathsf{Tr}}(|\Psi_S\rangle, |+^n\rangle)$. To that end, observe that:

$$\left(\sqrt{\frac{k}{n} + \frac{1-k/n}{2^k}} - \sqrt{\frac{1-k/n}{2^k}}\right)\frac{1}{\sqrt{2^k}} \le \frac{1}{\sqrt{2^{k+1}}}\sqrt{1 - \frac{k}{n}}. \tag{60}$$

from which we have that:

$$\left(1 + 2^{-(k+1)/2}\right)\sqrt{1 - \frac{k}{n}} \ge \langle +^n|\psi_S\rangle \ge \sqrt{1 - \frac{k}{n}}, \tag{61}$$

and[8] $\sqrt{\frac{k}{n}} \ge d_{\mathsf{Tr}}(\psi_0, \psi_S) \ge \sqrt{\frac{k}{n} - \frac{4}{2^{k/2}}}$. For $k \ge 2\log n$, $n \ge 5$ and $\tau \le \sqrt{\frac{2\log(n/4)}{n}}$, the planted biclique can be detected by at most $\binom{n}{k}$ Qstat$(\tau)$ queries. On the other hand, the $k$-planted biclique problem has $d_{\mathsf{TV}}(D, D_i) = \frac{k}{n}\left(1 - 2^{-k}\right)$ for all $D_i \in \mathcal{D}_D$, from which $d_{\mathsf{TV}}(D, D_0) = \frac{k}{n}(1 - 2^{-k}) < \frac{k}{n}$. It follows that:

$$\max_{\phi, |\phi| \le 1}\Pr_{D \sim \mathcal{D}}\left[|D[\phi] - D_0[\phi]| \ge 2\tau\right] \le \Pr_{D \sim \mathcal{D}}\max_{\phi, |\phi| \le 1}\left[|D[\phi] - D_0[\phi]| \ge 2\tau\right]$$
$$= \Pr_{D \sim \mathcal{D}}\left[d_{\mathsf{TV}}(D, D_0) \ge \tau\right]. \tag{62}$$

For $\tau = k/n$, we have $\Pr_{D \sim \mathcal{D}}\left[d_{\mathsf{TV}}(D, D_0) \ge \frac{k}{n}\right] = 0$, which means that the clique state is undetectable to a Stat$(\tau)$ algorithm. For $k \ge 2\log n$ and large enough $n$ ($n \gtrsim 72$), the statistical queries have better tolerance than the quantum queries. It follows that for $k \ge 2\log n$, $n \ge 72$ and $\tau = \sqrt{\frac{2\log(n/4)}{n}}$, the $k$-planted biclique problem cannot be solved by a Stat$(\tau)$ algorithm, but can be solved with an algorithm that can makes Qstat$(\tau)$ queries. $\square$

## 5.6 Hardness of Learning Approximate Designs

In this class we show that the classes of quantum states which form approximate 2-designs are hard to learn in the QSQ model.

**Theorem 13.** *Let $\mathcal{C}$ be an ensemble of states forming a $\gamma$-approximate $2$-design where $\gamma = O(2^{-n})$. Learning states from $\mathcal{C}$ with error $\le 1/3$ in trace distance requires $\Omega(\tau^2 \cdot 2^n)$ Qstat$(\tau)$ queries.*

*Proof.* We prove the theorem by showing that the variance of $\{\mathsf{Tr}[M\rho]\}_{\rho \in \mathcal{C}}$ for any such design must be exponentially small. By the definition of an approximate design, we have that

$$d_{\mathsf{Tr}}\left(\mathbb{E}_{\rho \sim \mathcal{C}}[\rho], \frac{1}{2^n}\mathbb{I}\right) \le \gamma, \quad d_{\mathsf{Tr}}\left(\mathbb{E}_{\rho \sim \mathcal{C}}[\rho^{\otimes 2}], \frac{1}{4^n + 2^n}(\mathbb{I} + \mathsf{SWAP})\right) \le \gamma, \tag{63}$$

where $\frac{1}{2^n}\mathbb{I}$ and $\frac{1}{4^n + 2^n}(\mathbb{I} + \mathsf{SWAP})$ are respectively the first and second moments of Haar distribution on pure states. For any observable $M$, by the definition of trace distance we have that

$$|\mathsf{Tr}[M(\frac{1}{2^n}\mathbb{I} - \mathbb{E}_{\rho \sim \mathcal{C}}[\rho])]| \le 2\gamma, \quad |\mathsf{Tr}[M(\frac{1}{4^n + 2^n}(\mathbb{I} + \mathsf{SWAP}) - \mathbb{E}_{\rho \sim \mathcal{C}}[\rho^{\otimes 2}])]| \le 2\gamma. \tag{64}$$

---

[8]Using $\sqrt{1 + 3 \times 2^{-(k+1)/2}} \ge 1 + 2^{-(k+1)/2}$ for all $k \ge 1$.

Thus, $\text{Var}_{\rho \sim \mathcal{C}}(\text{Tr}[M\rho]) \leq \text{Var}_{\rho \sim \mathcal{U}(2^n)}(\text{Tr}[M\rho]) + O(2^{-n})$. We now show that $\text{Var}_{\rho \sim \mathcal{U}(2^n)}(\text{Tr}[M\rho]) = O(2^{-n})$ for any $\|M\| \leq 1$.

$$\text{Var}_{\rho \sim \mathcal{U}(2^n)}(\text{Tr}[M\rho]) = \frac{1}{4^n + 2^n}\text{Tr}[M^{\otimes 2}(\mathbb{I} + \text{SWAP})] - \frac{1}{4^n}\text{Tr}[M]^2 \tag{65}$$

$$= \frac{1}{4^n + 2^n}\text{Tr}[M^2] - \frac{1}{2^n(4^n + 2^n)}\text{Tr}[M] , \tag{66}$$

where we have used that fact that $\text{Tr}[M^{\otimes 2}] = \text{Tr}[M]^2$ and $\text{Tr}[M^{\otimes 2}\text{SWAP}] = \text{Tr}[M^2]$. As $\|M\| \leq 1$ we have that $\text{Tr}[M^2] \leq 2^n$. Thus, taking $M$ to have $2^{n-1}$ eigenvalues equal to $+1$ and $2^{n-1}$ equal to $-1$ maximizes the variance yielding

$$\text{Var}_{\rho \sim \mathcal{C}}(\text{Tr}[M\rho]) \leq \text{Var}_{\rho \sim \mathcal{U}(2^n)}(\text{Tr}[M\rho]) + O(2^{-n}) \leq \frac{2^n}{4^n + 2^n} + O(2^{-n}) = O(2^{-n}) . \tag{67}$$

To invoke Theorem 4 and Lemma 1 we first note that all $\rho' \in \mathcal{C}$ are far from $\mathbb{E}_{\rho \sim \mathcal{C}}[\rho]$ in trace distance. This follows via triangle inequality:

$$d_{\text{Tr}}(\rho', \mathbb{E}_{\rho \sim \mathcal{C}}[\rho]) \geq d_{\text{Tr}}(\rho', \frac{1}{2^n}\mathbb{I}) - d_{\text{Tr}}(\frac{1}{2^n}\mathbb{I}, \mathbb{E}_{\rho \sim \mathcal{C}}[\rho]) = \frac{2^n - 1}{2^n} - O(2^{-n}).$$

Fix $\varepsilon = 1/3$. We assume that $\tau < \frac{1}{6}$ without loss of generality[9]. Thus, there is some $n_0$ such that for all $n \geq n_0$ we have that $d_{\text{Tr}}(\rho', \mathbb{E}_{\rho \sim \mathcal{C}}[\rho]) > 2(\tau + \varepsilon)$. Using Lemma 1 we thus have that learning states from $\mathcal{C}$ requires $\Omega(\tau^2 \cdot 2^n)$ Qstat queries. $\square$

# 6 Applications

## 6.1 Error mitigation

In this section, we show how to use our QSQ lower bound to resolve an open question posed by Quek et al. [11]. Therein the authors consider two forms of quantum error mitigation, which they call *strong* and *weak* error mitigation. We first describe these two models before stating our result.

**Definition 3** (Weak Error Mitigation). *An $(\varepsilon, \delta)$ weak error mitigation algorithm $\mathcal{A}$ takes an input a series of observables $\{O_1, \ldots, O_m\}$ satisfying $\|O_i\| \leq 1$ and outputs a set of values $\{\alpha_1, \ldots, \alpha_m\}$ such that with probability at least $1 - \delta$ we have that*

$$|\text{Tr}[O_i\rho] - \alpha_i| \leq \varepsilon . \tag{68}$$

**Definition 4** (Strong Error Mitigation). *An $(\varepsilon, \delta)$-strong error mitigation algorithm $\mathcal{A}$ outputs a bitstring $z$ sampled from a distribution $\mu$ such that, with probability at least $1 - \delta$, $d_{\text{TV}}(\mu, \mu_\rho) \leq \varepsilon$ where $\mu_\rho$ is the distribution on the computational basis induced by the state $\rho$.*

In [11] the authors show that strong error mitigation implies weak error mitigation for local observables. They then prove a partial converse and show that for a restricted family of observables weak error mitigation cannot recover strong error mitigation (for polynomial-sized inputs). The question of an unconditional separation is left open. Here we will show that Theorem **??** closes this open question and implies that weak error mitigation with polynomial numbers of observables does not suffice to recover strong error mitigation. First, note that by definition weak error mitigation outputs QSQ queries with tolerance $\tau = \varepsilon$. To match our notation, we will continue by using $\tau$ instead of $\varepsilon$. This is the equivalent of Lemma 5 of [11]. Second, we note that Theorem 5 of [11] still holds for our purposes. We repeat it here for completeness.

**Theorem 14.** *[11, Theorem 5] For a class of distributions $\mathcal{Q} = \{q_1, \ldots, q_k\}$ and $\varepsilon, \delta > 0$ there is an algorithm which takes $O(\frac{\log |\mathcal{Q}|}{\varepsilon^2})$ samples from a target distribution $p$ (not necessarily in $\mathcal{Q}$) and outputs a $q^* \in \mathcal{Q}$ such that*

$$d_{\text{TV}}(p, q^*) \leq 3\min_{i \in [k]} d_{\text{TV}}(p, q_i) + \varepsilon . \tag{69}$$

With these tools we can now prove a separation between strong and weak error mitigation.

---

[9]Again, these choices of constants are arbitrary. Choosing any $\varepsilon < 1/2$ is sufficient.

**Fact 7.** *For a distribution $p : \{0,1\}^n \to [0,1]$, let $|\psi_p\rangle = \sum_x \sqrt{p(x)}|x\rangle$. Suppose there exists an algorithm that makes $t$ Qstat queries and learns $p$ up to total variation distance $\varepsilon^2$, then there exists an algorithm that makes $t$ Qstat queries and learns $|\psi_p\rangle$ up to trace distance $\sqrt{2}\varepsilon$.*

*Proof.* By Fact 3, first observe that $d_{\mathsf{Tr}}(|\psi_p\rangle, |\psi_q\rangle)^2 \leq 2d_{\mathsf{TV}}(p,q)$ . Now the lemma statement follows by immediately: suppose there exists an algorithm that makes Qstat queries to $|\psi_p\rangle$ and outputs a $q$ such that $d_{\mathsf{TV}}(p,q) \leq \varepsilon^2$, then that implies that $d_{\mathsf{Tr}}(|\psi_p\rangle, |\psi_q\rangle) \leq \sqrt{2}\varepsilon$. $\qquad\square$

**Theorem 15.** *Let $\mathcal{A}$ be an algorithm that takes as inputs the estimates for weak error mitigation with $\tau = 1/\operatorname{poly}(n)$ and outputs $O(n^2)$ samples from some distribution $\mu$ such that $d_{\mathsf{TV}}(\mu, \mu_\rho) \leq 1/64$ with probability at least $0.99$. Then, $\mathcal{A}$ requires estimates of $\Omega(\tau^2 \cdot 2^{n/2})$ distinct observables.*

*Proof.* We show that such samples would give one the ability to learn quadratic polynomial states with polynomial QSQ queries, contradicting Theorem 5. First, we give a lemma connecting learning states of a specific form to learning their output distributions. Say that such an algorithm $\mathcal{A}$ does indeed exist. Then via Theorem 14, and noting that $\log|\mathcal{C}| = O(n^2)$, we can obtain a $\mu_{f^*}$ such that

$$d_{\mathsf{TV}}(\mu, \mu_{f^*}) \leq 3 \min_{f \in \mathcal{C}} d_{\mathsf{TV}}(\mu, \mu_f) + \frac{1}{64} . \tag{70}$$

By the assumption upon $\mu$ we further have that $d_{\mathsf{TV}}(\mu, \mu_{f^*}) < 1/16$. As $d_{\mathsf{TV}}(\mu_f, \mu_g) \geq \frac{1}{8}$ for $f \neq g$, we thus have, via triangle inequality, that $f^* = f$ and the true polynomial (and quantum state) can be recovered. This implies that the inputs to $\mathcal{A}$ could have been used a Qstat queries to solve the approximate state learning problem of Theorem 4. Thus, $\mathcal{A}$ requires $\Omega(\tau^2 \cdot 2^{n/2})$ distinct observables as inputs. $\qquad\square$

**Remark 2.** *For some forms of error mitigation it may be interesting to consider not just allowing the algorithm to query the circuit $U_\mathcal{C}$ but also modified circuits $U_{\mathcal{C}'}$. However this can be subsumed into the framework of weak error mitigation as given. To return an estimate of $O$ for $U_{\mathcal{C}'}$ the algorithm returns an estimate of $U_\mathcal{C} U_{\mathcal{C}'}^\dagger O U_{\mathcal{C}'} U_\mathcal{C}^\dagger$ from the original circuit.*

## 6.2 Learning distributions

In this section, we consider the following setup of statistical query learning that was considered in the work of [12]. Let $U$ be a unitary and consider the induced distribution $P_U$ on the computational basis, i.e.,

$$P_U(x) = \langle x|U|0^n\rangle^2.$$

In [12, 13] they considered learning algorithms that were given access to the following: for $\phi : \{0,1\}^n \to [-1,1]$ and $\tau \in [0,1]$,

$$\mathsf{Stat} : \phi, \tau \to \alpha_\phi \in \left[ \mathbb{E}_{x \sim P_U}[\phi(x)] + \tau, \mathbb{E}_{x \sim P_U}[\phi(x)] - \tau \right].$$

The goal of the learning algorithm is to learn $P_U$ up to total variational distance $\leq \varepsilon$ by making $\operatorname{poly}(n)$ many Stat queries each with tolerance $\tau = 1/\operatorname{poly}(n)$. Hinsche et al. [12] showed the hardness of learning the distribution $P_U$ when $U$ is a Clifford circuit of depth $\omega(\log n)$ and recently Neitner et al. [13] showed that if $U$ is a depth-$\Omega(n)$ circuit where each gate is picked from $U(4)$, then $P_U$ is not learnable using just Stat queries.

In this section we consider a stronger question. One can also just directly look at the quantum state $|\psi_U\rangle = U|0^n\rangle$ and ask how many Qstat queries of the form

$$\mathsf{Qstat} : M, \tau \to \alpha_M \in \left[ \langle\psi_U|M|\psi_U\rangle + \tau, \langle\psi_U|M|\psi_U\rangle - \tau \right].$$

suffice to learn $P_U$ upto small trace distance? Note that the learning model in [12, 13] is *a strict restriction* of this model, cause one could just consider $M = \sum_x \phi(x)|x\rangle\langle x|$, then

$$\langle\psi_U|M|\psi_U\rangle = \sum_x \phi(x)\langle x|U|0^n\rangle^2 = \sum_x \phi(x)P_U(x) = \mathbb{E}_{x \sim P_U}[\phi(x)],$$

which is precisely $\alpha_\phi$. To this end, we first generalize [12] in the following theorem.

**Theorem 16.** *For constant $\alpha \in (0,1)$, there is a family of $n$-qubit circuits of depth $d = (\log n)^{1/\alpha}$ and size $d^2$ that requires $2^{\Omega(d)}$ Qstat queries to learn the output distribution in the computational basis to error $\leq 0.00125$ in total variational distance. Lastly, note that these states can be prepared by performing Hadamard gates to obtain $\frac{1}{\sqrt{2^n}} \sum_x |x, 0\rangle$ followed by $O(n^2)$ tiffoli and CNOT gates, corresponding to the terms in the polynomial $x^\top A x$.*

*Proof.* Consider the padded states $|\psi_A\rangle \otimes |0\rangle^{\otimes k(n)}$, similar to those in Theorem 7 (here $\{|\psi_A\rangle = \frac{1}{\sqrt{2^n}} \sum_x |x, x^\top A x\rangle\}_A$). Using Fact 3 learning the output distributions of these states below total variational distance 0.00125 implies the existence of an algorithm learning the states up to trace distance 0.05. However, we know that doing so requires $2^{\Omega(d)}$ queries via Theorem 7. Thus, learning the output distributions requires at least $2^{\Omega(d)}$ Qstat queries as well. $\qquad\square$

We next prove a generalization of [13]. Before that, we need the following result.

**Theorem 17.** *[13, Theorem 36] There exists a $d = O(n)$ such that for any circuit depth $d' \geq d$ and any distribution $Q$ over $\{0,1\}^n$, we have that*

$$\Pr_{\psi \sim \mu_{d'}}[d_{\mathsf{TV}}(P_\psi, Q) \geq \frac{1}{225}] \geq 1 - O(2^{-n}).$$

**Theorem 18.** *Let $\mathcal{A}$ be an algorithm that satisfies the following: making $T$ many $\mathsf{Qstat}(\tau)$ queries, with probability $\geq 0.9$, $\mathcal{A}$ learns the output distributions of $O(n)$-depth random circuits, to error $< 1/225$ in total variational distance, then $T = \Omega(\tau^2 \cdot 2^n)$.*

*Proof.* This is a generalization of [13, Theorem 6] and follows by a similar analysis to their lower bound. For $d \geq 3.2(2 + \ln 2)n + \ln n$ we have that the uniform distribution over depth $d$ random circuits is a $2^{-n}$ approximate 2-design [13]. Like we saw in the proof of Theorem 13, for every observable $\|M\| \leq 1$, we have that $\mathsf{Var}_{\rho \sim \mathcal{C}}(\mathsf{Tr}[M\rho]) = O(2^{-n})$. We proceed with the same adversarial lower bound. Upon making a $\mathsf{Qstat}$ query with observable $M$, the adversary responses with $\mathbb{E}_{\rho \sim \mathcal{C}}[\mathsf{Tr}[M\rho]]$. Using Chebyshev's inequality,

$$\Pr_{\rho \sim \mathcal{C}}\left[|\mathsf{Tr}[M\rho] - \mathsf{Tr}[M \mathop{\mathbb{E}}_{\rho \sim \mathcal{C}}[\rho]]| > \tau\right] = \tau^{-2} \cdot 2^{-n}. \tag{71}$$

While the proof could continue using Theorem 4 (by reducing to the many-vs-one decision problem), it is more direct to note that the above inequality implies that every deterministic algorithm cannot identify the correct $\rho \in \mathcal{C}$ for a large fraction of states in $\mathcal{C}$. In particular, say $\mathcal{A}$ is a deterministic algorithm that outputs an estimate of a distribution $Q$ such that $d_{\mathsf{TV}}(P_\psi, Q) < 1/225$ and uses at most $t$ $\mathsf{Qstat}(\tau)$ queries. Using Eq. (71), there is a fraction of $\mathcal{C}'$ of measure at least $1 - O(t \cdot \tau^{-2} \cdot 2^{-n})$ that are consistent with $\mathsf{Tr}[M_i \mathbb{E}_{\rho \sim \mathcal{C}}[\rho]]$ for the $\mathsf{Qstat}(\tau)$ queries $\{M_1, \ldots, M_t\}$ made by $\mathcal{A}$. Since $\mathcal{A}$ is deterministic, it must output the same distribution $Q$ for all $\rho \in \mathcal{C}'$. We now use Theorem 17 to claim that there is a large set of states that are both consistant with $\mathsf{Tr}[M_i \mathbb{E}_{\rho \sim \mathcal{C}}[\rho]]$ and far from $Q$ in total variational distance.

$$\Pr_{\rho \sim \mathcal{C}}[\rho \in \mathcal{C}' \wedge d_{\mathsf{TV}}(P_\rho, Q) \geq 1/225] = 1 - \Pr_{\rho \sim \mathcal{C}}[\rho \notin \mathcal{C}' \vee d_{\mathsf{TV}}(P_\rho, Q) < 1/225] \tag{72}$$

$$\geq 1 - O(t \cdot \tau^{-2} \cdot 2^{-n}), \tag{73}$$

where the inequality follows from the union bound, Theorem 17, and the concentration of measure shown above. Thus, there is a set $\mathcal{C}''$ of measure at least $1 - O(t \cdot \tau^{-2} \cdot 2^{-n})$ that is both consistent with $\mathsf{Tr}[M_i \mathbb{E}_{\rho \sim \mathcal{C}}[\rho]]$ for all queries $M_i$ and also $d_{\mathsf{TV}}(P_\rho, Q) \geq 1/225$ for all $\rho \in \mathcal{C}''$. Upon the input of a $\rho \in \mathcal{C}''$, $\mathcal{A}$ fails to provide a distribution $Q$ such that $d_{\mathsf{TV}}(P_\rho, Q) < 1/225$. Note that the measure of $\mathcal{C}''$ is at least 0.99 for all $n$ sufficiently large and $t \in \mathrm{poly}(n)$. Via Yao's Principle, thus any randomized algorithm using $t \in \mathrm{poly}(n)$ Qstat queries must fail with probability at least 0.99 for $n$ sufficiently large. Further, for the measure of $\mathcal{C}''$ to be at most 0.01 for all $n$, the algorithm must use $t = \Omega(\tau^2 \cdot 2^n)$ Qstat queries. $\qquad\square$