# OpenReview forum: "On the Role of Entanglement and Statistics in Learning"
_NeurIPS.cc/2023/Conference — NeurIPS 2023 poster_

### Official Review · Reviewer_UDTy · 2023-06-13

**Soundness:** 4 excellent
**Presentation:** 2 fair
**Contribution:** 2 fair
**Rating:** 6
**Confidence:** 4

**Summary:**

This paper studies quantum learning theory, in particular the relationship between the quantum version of PAC learning and the quantum version of the statistical query (QSQ) model, as well as their connection to other considerations in quantum computing, such as entangled measurements and separable measurements. Specifically, the authors proved that

1. For learning Boolean concept classes, the entangled and separable sample complexity are polynomially related (at most quadratic power).

2. There exists a concept class with an exponential separation between quantum PAC learning with classification noise and QSQ learning.

Along this, the authors also proposed novel technical tools of quantum statistical query dimension, and these technical results are applied to problems ranging from states learning, distribution learning, and specific problems in quantum computing such as shadow tomography, error mitigation, etc.

**Strengths:**

This paper is technically very solid and is able to prove a series of new results in quantum learning theory. PAC learning and SQ learning models are both important concepts in classical learning theory, and it’s nice to see that the authors are able to generalize them to the quantum domain, connect them to natural definitions in quantum, and prove separation results between these concepts. It’s also nice to see that the authors are able to find wide applications in quantum learning theory.

**Weaknesses:**

From my perspective, the most notable weakness of this works is its interest to the general machine learning community. There are theory papers in NeurIPS each year, many of which are very interesting and well received by NeurIPS audiences, but I’m afraid that this one falls into too much on the theory side and might be of limited interest to the NeurIPS community. In general, the topics of PAC learning and SQ model are more of theoretical interest and target more on theoretical computer science. The quantum version further delves into those directions and bring into definitions that do not exist in classical machine learning, such as entangled measurements/separable measurements, shadow tomography, etc. As another evidence, the references in the main body contain literally 0 paper coming from top-tier machine learning conferences targeting at general audiences, including, NeurIPS, ICML, ICLR, AAAI, etc. Instead, there are many top-tier theory papers at STOC/FOCS, Journal of the ACM, and top-tier physics journals. In general, I believe that this paper can be quite competitive in top-tier theoretical computer science venues or quantum physics venues, but is out of scope for NeurIPS due to lack of insights for up-to-date trends in current machine learning research.

As a minor point of weaknesses, I think the references can be better presented. First, I find it a bit hard to locate references: there are quite a few big brackets citing >5 papers at the same time, and it’s very difficult to determine which paper talks about which topic. For instance, in Page 1, “There have already been many theoretical proposals for quantum algorithms providing speedups for practically relevant ML tasks such as clustering, recommendation systems, linear algebra, convex optimization, SVMs, kernel-based methods, topological data analysis [34, 14, 9, 42, 32, 26, 35, 23, 49, 46]” can be better written as … such as clustering [xx], recommendation systems [xx], linear algebra [xx], convex optimization [xx], SVMs [xx], kernel-based methods [xx], “and” topological data analysis [xx].

Actually, I think here the authors lost some quantum computing papers accepted by past NeurIPS/ICML conferences that provide quantum speedup for solving machine learning problems, such as Kapoor et al. (https://proceedings.neurips.cc/paper/2016/hash/d47268e9db2e9aa3827bba3afb7ff94a-Abstract.html) and Li et al. (http://proceedings.mlr.press/v97/li19b.html) for classification, Arunachalam and Maity (https://proceedings.mlr.press/v119/arunachalam20a.html) for boosting, Childs et al. (https://proceedings.neurips.cc/paper_files/paper/2022/hash/933e953353c25ec70477ef28e45a2dcc-Abstract-Conference.html) for logconcave sampling, etc.

**Questions:**

Can the techniques in this paper be applied to prove separation results between quantum and classical PAC learning or SQ models? As far as I see, this paper studies purely quantum learning concepts and their relationships. It would be helpful to deliver more conceptual message about the difference between quantum and classical learning, which may be of general interest (this is also the storyline set at the beginning of intro, but quickly drives into purely quantum things).

**Limitations:**

N/A – this work is purely theoretical.

---

> ### Author Rebuttal · Authors · 2023-08-07
>
> Thank you for your comments!
>
> Regarding your question about separating quantum vs. classical techniques: There are known separations between classical and quantum PAC for the distribution-dependent setting, for example for DNF formulas [Bshouty, Nader H., Jackson, Jeffrey C.: Learning DNF Over the Uniform Distribution Using a Quantum Example Oracle; COLT '95]. Similarly, there are separations between classical SQ and quantum SQ witnessed by the parity functions [Arunachalam, Grilo, Yuen: Quantum Statistical Query Learning; CoRR]. Prior to the submitted work, no separation was known between quantum SQ and quantum PAC, which is why we focus on that. Based on your remark, we will improve the discussion of prior results in the introduction and focus on the delivery of the conceptual message to the broader community. Thank you for pointing that out!
>
> We value your comment regarding references and will update the introduction accordingly with references to these results in the revision. We will add references and improve their presentation according to your suggestion.
>
> On a higher level, our results provide an insight into where to look for quantum advantages in machine learning and what resources this requires—something we feel is of broader interest in the ML community. Thus, it is interesting to characterize the limitations of QSQ, a natural generalization of SQ that models near-term capabilities in quantum hardware. Our work can be viewed as a provable separation between QML with near term hardware and QML with more sophisticated hardware, by introducing techniques to understand the QSQ model better.
>
>  On the classical side, the SQ model is intimately linked to (local) differential privacy and optimization, in particular, theoretical and practical algorithms fall into the SQ framework [Feldman, Ghazi; On the Power of Learning from k-Wise Queries; ITCS '17]. Theoretical results on PAC/SQ learning and quantum learning theory [Arunachalam, Quek, Smolin: Private Learning Implies Quantum Stability; NeurIPS ‘21] have been published earlier in NeurIPS. We believe that understanding their counterpart in the quantum setting is of interest for the broader ML community, especially those parts of it that have interest in quantum, and that NeurIPS is an appropriate venue for our work.
>
> Based on your recommendation, we will improve the discussion of quantum vs classical learning, to set the context for the typical audience at NeurIPS. Due to the lack of space for the NeurIPS submission we didn't delve into this to sufficient depth, but will implement it in the revision.

---

> > ### Comment · Reviewer_UDTy · 2023-08-10
> > **Acknowledgement**
> >
> > I would like to thank the authors for the detailed rebuttal, which makes the storyline more complete from my perspective. It would be helpful if the discussions in the rebuttal can be combined into the final version of the paper, since an additional page will be given.

---

### Official Review · Reviewer_JWWq · 2023-06-27

**Soundness:** 3 good
**Presentation:** 3 good
**Contribution:** 3 good
**Rating:** 7
**Confidence:** 4

**Summary:**

This paper studies the power of different quantum machine learning models for learning Boolean functions, namely quantum PAC-learning (QPAC) with entangled measurements, QPAC with separable measurements, and quantum statistical query (QSQ). It has two main results. First, it shows that QPAC with entanglement measurements is not more powerful than separable measurements in this task. More specifically, to *exactly* learn an $n$-bit Boolean function class, every learning algorithm using entangled measurements with $T$ copies of the quantum sample $|\psi_c\rangle=2^{-n/2}\sum_x |x,c(x)\rangle$ can be transformed into a learning algorithm using just separable measurements with O(nT^2) samples. Second, it provides an exponential separation between QPAC with noise and QSQ. In the QSQ model, the learner can query the oracle with any observable $M$ (implementable with $poly(n)$ gates) and obtain an estimate of $tr[M\rho]$ within $1/poly(n)$-additive error. It constructs a concept class of $n$-bit Boolean functions that is QPAC learnable with $\eta$-classification noise (i.e., $|\psi_c\rangle=2^{-n/2}\sum_x \sqrt{1-\eta}|x,c(x)\rangle+\sqrt{\eta}|x,1-c(x)\rangle$) in time $poly(n, 1/(1 − 2\eta))$, whereas every QSQ learner requires $2^{\Omega(n)}$ queries. Furthermore, it also provides several applications in shadow tomography, testing purity of quantum state, error mitigation, and learning output distributions of quantum circuits.

**Strengths:**

Understanding the power of different quantum learning models is an important research question in quantum learning theory. The results of this paper are quite solid. More specifically, for the first main result about the relation between entangled measurements and separable measurements, prior to this work, we only knew that they are exponentially separated when learning quantum states. This paper focuses on a smaller concept class and shows that they are polynomially related, which is quite surprising. For the second result about measurement statistics, compared to the classical result that separates PAC from SQ due to Blum et al., the construction in this paper is based on degree-2 polynomials, which are more natural. And the proof of this result provides a novel approach to lower bound the complexity of QSQ via the quantum statistical dimension, which is the main technical contribution of this paper. The applications also improve over prior works in several aspects. Furthermore, this paper is well-structured, and most proofs are mathematically sound to me.

**Weaknesses:**

This paper may seem quite difficult to follow for people not working on quantum computing. Also, the proof overview section is too technical, and more intuition should be given.

The relation between entangled measurements and separable measurements is $O(nT^2)$, which seems to be not tight. (It has already been pointed out in the paper)

**Questions:**

* In Sec. 3.2.1, it could be better to provide the classical definition of the statistical dimension and compare it to the quantum one.

* This paper considers proper learning. What about quantum improper learning? The input is still the quantum sample state for some unknown Boolean function, but the goal is just to output some quantum state close to the given quantum sample state. Are the techniques in this paper still apply to this setting?

**Limitations:**

The limitations are stated. Potential negative societal impact does not apply here.

---

> ### Author Rebuttal · Authors · 2023-08-07
>
> Thank you for your very interesting questions/comments.
>
> Indeed, our bound on $O(n T^2)$ is suboptimal, but in our main theorem statement we have a slightly combinatorial parameter $\eta$ and we show a bound of $O(n T \eta)$ which we show is tight for a certain concept class. We suspect that the right relation should be $O(nT)$ and leave it for future work.
>
> Regarding proper versus improper: our results do $\textit{not}$ assume that the learning algorithms are proper. In particular, our lemma bounding learning complexity with decision complexity holds for both proper and improper learners. Then, of course, for decision problems there is no concept of proper/improper learning. In our revision we will make it clear that our lower bound holds for even improper learners.

---

> > ### Comment · Reviewer_JWWq · 2023-08-14
> >
> > I thank the authors for their response. I keep my score.

---

### Official Review · Reviewer_gCmC · 2023-07-05

**Soundness:** 4 excellent
**Presentation:** 3 good
**Contribution:** 4 excellent
**Rating:** 8
**Confidence:** 4

**Summary:**

This submission investigates the relationship between learning models with access to entangled measurements, separable measurements, and statistical measurements in the quantum statistical query (QSQ) model. The authors make several notable contributions: they establish the polynomial relationship between the sample complexity of entangled and separable measurements for learning Boolean concept classes, demonstrate an exponential separation between quantum PAC learning with classification noise and QSQ learning, introduce the concept of quantum statistical query dimension (QSD) to provide lower bounds on QSQ learning, prove exponential QSQ lower bounds for various tasks, show an unconditional separation between weak and strong error mitigation, and derive lower bounds for learning distributions in the QSQ model. These contributions advance our understanding of quantum learning theory and have implications for the development of quantum learning algorithms.






**Strengths:**

The paper successfully addresses multiple open questions regarding the capabilities of diverse quantum learning models. The insights gained from the obtained results offer a valuable understanding of the potentials and limitations of near-term quantum machines in comparison to fault-tolerant quantum computers. Additionally, the authors leverage the findings from the QSQ model to enrich our comprehension of crucial aspects in the field of near-term quantum machines, such as error mitigation and distribution learning. The theoretical contributions made in this work play a vital role in advancing quantum learning theory and provide practical guidance for developing and optimizing quantum algorithms on both current and future quantum hardware.






**Weaknesses:**

According to the whole main text, I did not find the main weakness of the submission While I did not review the entire proof, the portion I examined did not reveal any apparent issues. However, I did notice some minor concerns such as typos and incorrect notations. I suggest that the authors thoroughly examine their work to address these issues, ensuring a self-consistent and easily understandable manuscript. For instance:
1. Line 346, 'a important'
2. Line 375, '?, (ii)'
3. Line 41, Supplementary: '$\sum_{x\in D_0}$'
4. Line 44, Supplementary: '$I(D_0(x)\leq D(x))$'
5. Line 123, Supplementary: 'a n-bit'
6. Line 602, Supplementary: 'Theorem ??'

**Questions:**

It is recommended that the authors address the issues outlined in the Weakness section.






**Limitations:**

The authors have acknowledged the limitation by discussing open questions in the Discussion section.

---

> ### Author Rebuttal · Authors · 2023-08-07
>
> Thank you for your comments!
>
> We will definitely address these and also look over the document for the next revision.

---

### Official Review · Reviewer_AabW · 2023-07-27

**Soundness:** 4 excellent
**Presentation:** 3 good
**Contribution:** 3 good
**Rating:** 6
**Confidence:** 3

**Summary:**

This paper considers the task of learning an unknown concept class with quantum accesses. In particular, the authors make a comprehensive comparison of the settings with entangled measurements, separable measurements, and statistical measurements in the quantum statistical query (QSQ) model, respectively, and showed that entangled measurements are at most polynomially more powerful than separated measurements, which are exponentially more powerful than QSQ learning. Notably, this second separation is a quantum analog of the classical result separating between classical SQ learning from classical PAC learning. The authors also discuss possible extensions and applications of their result.

**Strengths:**

This paper provides a clear answer on the possible role of entanglement the quantum PAC learning setting, and establishes a distinct separation between quantum PAC learning and QSQ learning, which provides an interesting conceptual message. The technical contribution of this paper is solid, and the presentation is clear.

**Weaknesses:**

The two main results of this paper are not closely related from my perspective.

**Questions:**

Following my previous point, could you please elaborate more on the connection between the two settings?

---

> ### Author Rebuttal · Authors · 2023-08-07
>
> Thanks for your comments.
>
> While the proof techniques seem very different in both these results, we believe that both the results are related within the theme "need for entanglement in learning". Previously, it was known that QSQ measurements >= Sep measurements >= Ent measurements for all learning tasks. Our work aims to solidify the understanding of the power of the three models and show that Sep and Ent are polynomially related and QSQ measurements are exponentially weaker than Ent measurements. In our revision, we will keep your comment in mind to make our paper look more cohesive.
>
> Thank you very much for pointing this out!

---

> > ### Comment · Reviewer_AabW · 2023-08-14
> >
> > I would like to thank the authors for the explanation. I remain my rating.

---

### Decision · Program_Chairs · 2023-09-21

**Decision:**

Accept (poster)

**Comment:**

Understanding the power of different quantum learning models is an important research question in quantum learning theory. This paper has made solid contributions to this research frontier. In particular, it studies the relationship between the quantum version of PAC learning and the quantum version of the statistical query (QSQ) model, as well as the difference between entangled and separable measurements. The paper establishes an exponential separation in the former while demonstrating a polynomial relation in the latter.  We wanted to ask the authors to include more discussions (e.g., from the rebuttal) on the connection of the learning theory results to the machine learning community, as well as the connection between the two main results in the revision.